# Extrinsically microporous polymer membranes derived from thermally cross-linked perfluorinated aryl-ether-free polymers for gas separation

Ju Ho Shin [1], Hyun Jung Yu[1], Jiyoon Jung[2], Heseong An[3], Jung Hoon Park[4], Albert S. Lee [2,5] ✉ & Jong Suk Lee [1,6,7] ✉

State-of-the-art membranes derived from polymers of intrinsic microporosity offer promising alternatives to energy-intensive, thermally driven separation techniques but often suffer from reduced performance under condensable gases or physical aging. Here, extrinsically microporous polymer membranes (EMPMs) are introduced as a distinct class of microporous membranes, fabricated from perfluorinated aryl-ether-free aromatic polymers via defluorination-induced thermal cross-linking. This process generates extrinsic micropores, increases intersegmental distances, and significantly enhances gas permeability. EMPMs exhibit a Brunauer-Emmett-Teller surface area of 552 $m^2 g^{-1}$ and demonstrate exceptional plasticization resistance under equimolar $CO_2/CH_4$ mixed gas at pressures up to 40 bar. $CO_2$ permeability increases from 280 to 12,000 Barrer at 1 bar and 35 °C, while $CO_2/N_2$ selectivity reaches 46 at −20 °C, surpassing the 2019 polymeric upper bound. Furthermore, extrinsically microporous hollow fiber membranes prepared via dip-coating achieve a $CO_2$ permeance of 2174 gas permeation units and $CO_2/N_2$ selectivity of 30 at −20 °C, highlighting their industrial relevance. This study establishes a scalable method for fabricating high-performance microporous polymeric membranes with exceptional stability for sustainable energy and environmental applications.

Chemical processes often require thermal or non-thermal molecular separation in downstream stages to produce high-purity chemicals, with thermal separation methods being the most common[1]. However, thermal processes that involve vaporization are highly energy-intensive, underscoring the need to transition to pressure-driven alternatives to reduce the carbon footprint[2]. Membrane separation

presents a promising solution in this regard. For instance, reverse osmosis membranes can reduce energy consumption by up to 90% in seawater desalination compared to distillation[3]. Polymer membranes, particularly, are appealing due to their high scalability and cost efficiency. Widely recognized commercial polymers like cellulose acetate and polysulfone exhibit moderate $CO_2/CH_4$ selectivity (21 and 22,

[1]Department of Chemical and Biomolecular Engineering, Sogang University, 35 Baekbeom-ro, Mapo-gu, Seoul 04107, Republic of Korea. [2]Materials Architecturing Research Center, Korea Institute of Science and Technology, 5 Hwarang-ro 14-gil, Seongbuk-gu, Seoul 02792, Republic of Korea. [3]Department of Chemical Engineering, Sunchon National University, 255 Jungang-ro, Suncheon-si, Jeollanam-do 57922, Republic of Korea. [4]Department of Chemical and Biochemical Engineering, Dongguk University, 30 Pildong-ro 1gil, Jung-gu, Seoul 04620, Republic of Korea. [5]Convergence Research Center for Solutions to Electromagnetic Interference in Future-Mobility, Korea Institute of Science and Technology (KIST), 5 Hwarang-ro 14gil, Seongbuk-gu, Seoul 02792, Republic of Korea. [6]Institute of Energy and Environmental Technology, Sogang University, 35 Baekbeom-ro, Mapo-gu, Seoul 04107, Republic of Korea. [7]Institute of Emergent Materials, Sogang University, 35 Baekbeom-ro, Mapo-gu, Seoul 04107, Republic of Korea. ✉e-mail: aslee@kist.re.kr; jongslee@sogang.ac.kr

respectively) but show low $CO_2$ permeability (8.9 Barrer and 5.6 Barrer, respectively)[4]. Polyimides have also been employed in gas separation, but their performance has been unsatisfactory ($CO_2$ permeability of 7−20 Barrer and $CO_2/CH_4$ selectivity of 58−102)[5]. In addition to the well-established trade-offs between permeability and selectivity in membranes, major challenges in gas separation using polymeric membranes include plasticization susceptibility and physical aging, both of which limit overall performance. Consequently, membrane material science has been a crucial topic in polymer research for several decades, with ongoing efforts to develop high-performance polymers for membrane applications[6].

Microporous polymers, such as polymers of intrinsic microporosity (PIM), thermally rearranged (TR) polymers, and Tröger's base polymers, have emerged as promising alternatives to conventional polymer membranes, providing significantly improved permeability in membranes[7-9]. In particular, PIMs represent a class of microporous polymers featuring fused ring structures with contortion sites along the backbone[7], resulting in frustrated chain packing in the solid state and the creation of permanent interconnected microporosity. This feature leads to exceptional permeability compared to traditional polymers. However, they are susceptible to selectivity loss due to plasticization when exposed to condensable gases at high feed pressures and a rapid decrease in permeability induced by physical aging[10]. Efforts to enhance their resistance to plasticization have primarily centered on thermal cross-linking under an inert gas atmosphere[11] and thermo-oxidative cross-linking[12]. Unfortunately, cross-linked membranes often exhibit lower permeability than the original microporous membranes due to reduced free volumes after cross-linking. More recent studies have also reported success in improving plasticization resistance by incorporating bulky and rigid moieties, such as fluorinated side chains[13] or triptycene units[14]. Despite increasing regulatory scrutiny over the persistence of fluorinated compounds, certain fluorinated polymers remain indispensable in membrane research due to their superior chemical resistance and thermal stability, which are difficult to achieve with non-fluorinated analogs.

This study presents a strategy for fabricating highly permeable, plasticization-resistant microporous polymer membranes derived from fluorinated aromatic glassy polymers, termed extrinsically microporous polymeric membranes (EMPMs). The approach leverages a thermally activated defluorination mechanism that generates free radical sites, which undergo inter-chain cross-linking, leading to the formation of a microporous polymer network with reduced residual fluorine content (Fig. 1a). Comprehensive thermal and spectroscopic analyses confirm the evolution of fluorinated fragments, the formation of aryl-aryl linkages, and structural rearrangements that give rise to a highly interconnected free volume architecture. These EMPMs demonstrate exceptional gas permeability and $CO_2$ separation performance, particularly under sub-ambient conditions, along with remarkable long-term stability and plasticization resistance. Furthermore, the membranes can be readily processed into asymmetric hollow fiber configurations, highlighting their scalability and practical potential. Overall, this work presents a promising pathway toward high-performance membrane materials, contributing to the advancement of sustainable gas separation technologies.

## Results

### Defluorination-driven fabrication of EMPMs from aryl-ether-free FAPs

Two aryl-ether-free fluorinated aromatic polymers (FAPs) were synthesized as representative examples via superacid-catalyzed poly-hydroxyalkylation between perfluoroacetophenone (or trifluoroacetophenone) and *para*-terphenyl, activated using tri-fluoromethanesulfonic acid (TFSA)[15]. This polymerization method enables the creation of a series of linear, high-molecular-weight

polymers, with structural variations based on different fluorinated ketones and multiring aromatic hydrocarbons. The chemical structures of the synthesized *para*-terphenyl trifluoroacetophenone (*p*TPTFA) and *para*-terphenyl perfluoroacetophenone (*p*TPPFA) were confirmed using the [1]H NMR spectroscopy (Supplementary Fig. 1), and their molecular weight, glass transition temperature, density, fractional free volume, and *d*-spacing are summarized in Supplementary Table 1. These FAPs exhibit rigid, heteroatom-bond-free aromatic backbones, imparting high thermal stability. Notably, polymers with pendant phenyl and pentafluorophenyl groups were chosen to investigate the effects of thermally activated defluorination-induced cross-linking on polymer structure (Fig. 1b), demonstrating their significant potential for high-performance membrane applications.

While the pristine membranes exhibit gas transport properties comparable to previously reported fluorinated polyimides, defluorination-induced thermal cross-linking significantly enhances their gas permeability. During thermal treatment at 400 °C to 500 °C under argon flow, fluorinated groups detach from the polymer backbone, creating reactive sites that facilitate interchain cross-linking. This cross-linking occurs primarily through the $sp^2$C−$sp^2$C bonds, with potential contributions from $sp^3$C linkages[16], leading to the formation of a carbon-like opaque film (Fig. 1c). Biphenyl linkages are likely formed through the coupling of aryl radicals, while trifluoromethyl radicals may recombine with phenyl radicals or with each other. This inter-chain cross-linking restricts the rotational freedom of the polymer backbone, leading to the formation of a permanent microporous structure that facilitates gas transport. This contrasts with previously reported methods, such as decarboxylation[17], debromination[18], and defluorination of 6FDA-derived polyimides[19], which primarily enlarge interchain spacing, resulting in only negligible increases in microporosity. Exposure to higher temperatures promotes aromatization, followed by fragmentation, which ultimately generates short, rigid carbon strands (Supplementary Fig. 2).

### Thermal stability and defluorination behavior of FAPs

To evaluate the thermal stability of FAPs, thermogravimetric analysis (TGA) was conducted to monitor the weight loss of *p*TPTFA and *p*TPPFA during thermal treatment in an $N_2$ atmosphere (Fig. 2a). For *p*TPPFA, the initial weight loss occurred between 350 °C and 500 °C, with a total weight loss of 10.3% at 500 °C. A subsequent degradation stage, attributed to the decomposition of the main chain, was observed between 500 °C and 550 °C. In contrast, *p*TPTFA exhibited the onset of thermal degradation at 480 °C, with a single major weight-loss event at 530 °C, likely associated with the degradation of $CF_3$ groups and the polymer backbone. To further analyze the evolution of fluorinated fragments during *p*TPPFA thermal treatment, simultaneous thermal analysis coupled with mass spectroscopy (STA-MS) was performed (Fig. 2b). Above 400 °C, *p*TPPFA released pendent fragments such as fluorine (m/z = 19), $HCF_2$ (m/z = 51), $CF_3$ (m/z = 69), and $C_6F_5$ (m/z = 168). Importantly, no decomposition of the main *para*-terphenyl backbone was detected up to 450 °C, as corroborated by TG-GC/MS data (Supplementary Fig. 3 and 4). In contrast, *p*TPTFA showed the removal of $CHF_2$ and $C_6H_5$ groups at higher temperatures (>510 °C), followed by carbonization due to $CF_3$ loss (Supplementary Figs. 5−7). These results highlight the thermal instability of the pentafluorophenyl group. Despite the higher bond dissociation energy (BDE) of C−F bonds compared to C−C bonds (485 vs. 348 kJ mol⁻¹), TG-MS results suggest that fluorine atoms in the pentafluorophenyl moiety may form highly reactive radicals upon thermal activation[16]. This observation aligns with earlier thermodynamic work by Cox[20] et al. who reported that the heat of formation of hexafluorobenzene deviates by approximately 159 kJ mol⁻¹ (38 kcal mol⁻¹) from values predicted by bond additivity, indicating that electronic repulsion within the fully fluorinated aromatic ring reduces its thermodynamic stability.

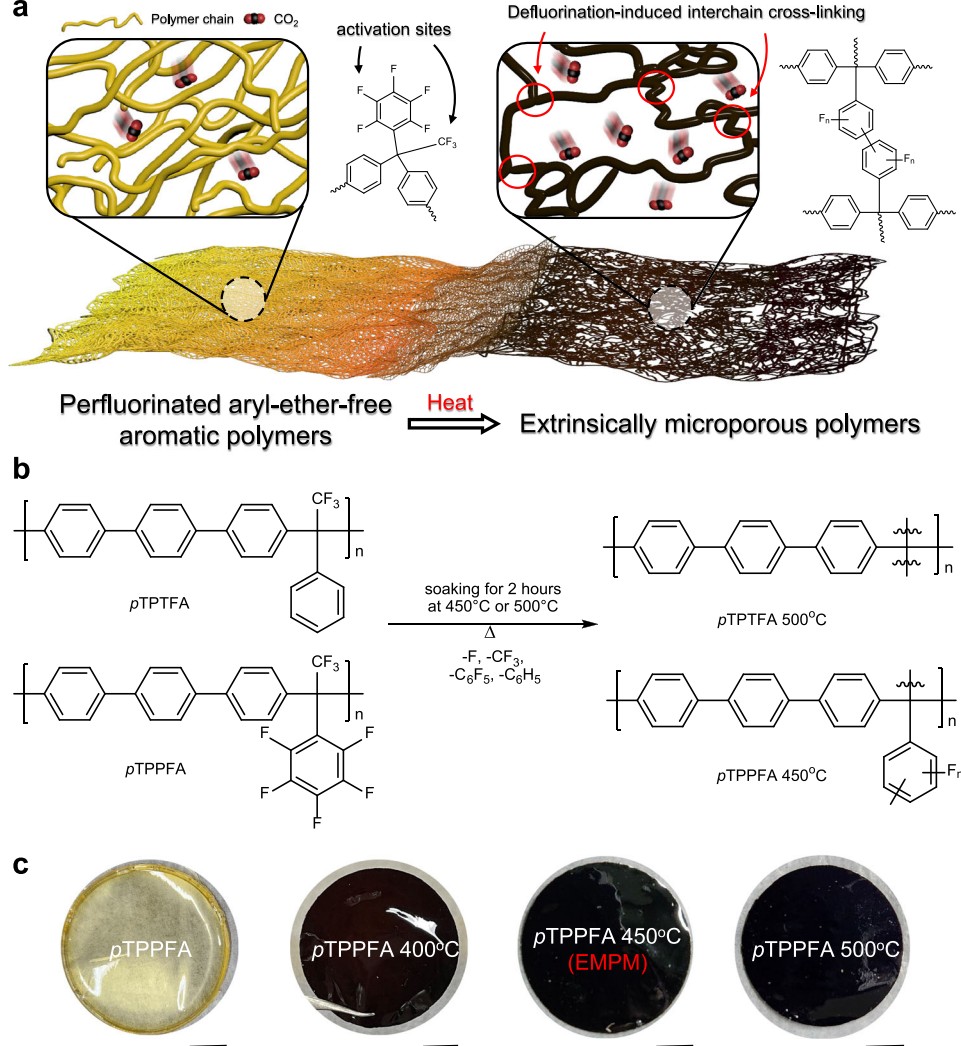

**Fig. 1 | Schematic representation of the defluorination and cross-linking mechanisms in fluorinated aromatic polymers. a** Illustration of the transformation into extrinsically microporous polymeric membranes (EMPMs). Thermal activation induces defluorination of pentafluorophenyl groups, generating reactive radicals that facilitate inter-chain cross-linking, resulting in a microporous structure. **b** Chemical structures of *p*TPTFA and *p*TPPFA, along with hypothetical fluorinated aromatic polymer strands after thermal treatment, highlighting structural rearrangements involving inter-chain radical coupling resulting from defluorination. **c** Optical images of pristine *p*TPPFA and thermally treated *p*TPPFA membranes, showing visual changes observed at 400 °C, 450 °C, and 500 °C. The diameter of each film is 45 mm, except for *p*TPPFA 500 °C (42 mm). Scale bar in all panels: 10 mm.

Such destabilization facilitates defluorination and may promote the formation of aryl radicals, ultimately driving thermally induced cross-linking.

### Defluorination-induced cross-linking and structural evolution of EMP membranes

Thermally treated membranes derived from *p*TPPFA and *p*TPTFA were prepared at soaking temperatures of 400 °C, 450 °C, and 500 °C, and are designated as *p*TPPFA *x* °C or *p*TPTFA *x* °C, where *x* indicates the temperature. Raman spectra of *p*TPPFA and its thermally treated forms confirm defluorination without carbonization between 400 °C and 450 °C (Fig. 2c), as evidenced by the absence of the G (1500–1600 cm⁻¹) and D (1300–1400 cm⁻¹) bands. The G band corresponds to the $E_{2g}$ symmetry stretching vibration mode of $sp^2$-hybridized graphitic carbon, while the D band is associated with the $A_{1g}$ breathing mode, arising from structural disorder in aromatic domains[21]. The intensities of the D and G bands increased in the *p*TPPFA 500 °C and 550 °C samples, indicating progressive formation of a carbon molecular sieve (CMS)[22,23]. Differential scanning calorimetry (DSC) revealed exothermic peaks in the ranges of 350–400 °C

and 400–480 °C, consistent with the thermal degradation of fluorinated groups (Supplementary Figs. 8–12). Modulated DSC (MDSC®) data suggested that carbonization follows defluorination beyond 480 °C. Additionally, no glass transition temperature was observed in the thermally treated membranes, indicating highly restricted polymer chain mobility. The mass loss during isothermal treatments increased with temperature (10.2% at 400 °C, 14.8% at 450 °C, and 22.5% at 500 °C, Supplementary Fig. 13). Aryl-ether- or heteroatom-linked polymers, such as cellulose, polyimide, and polybenzodioxane, are often used as polymer precursors for CMS membranes[22–24]. The initial stage of aryl-ether- or heteroatom-linked polymer chain scission leads to the formation of rigid carbon structures at elevated decomposition temperatures (>450 °C), resulting in packing imperfections that create micropores[25]. However, in *p*TPPFA, early-stage defluorination likely generated reactive aryl radicals, enabling cross-linking. The electron-withdrawing nature of fluorine destabilized the pendent group, sparing the backbone from carbon strand formation. These findings suggest that pentafluorophenyl groups primarily drive cross-linking in *p*TPPFA. The thermally treated membranes (thickness: $60 \pm 5$ μm) retained their structural integrity. Nano-indentation results

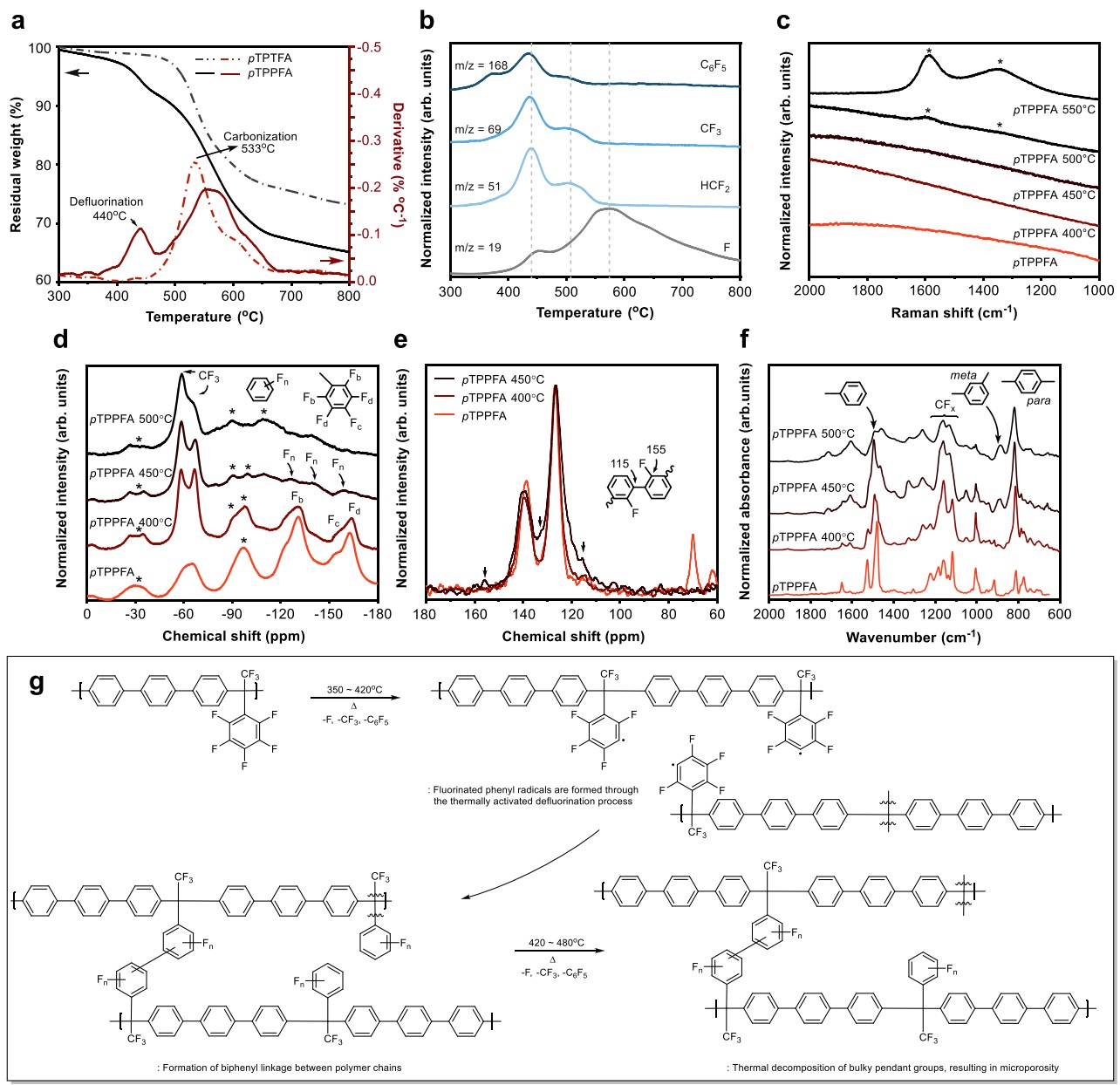

**Fig. 2 | Characterization of fluorinated aromatic polymers and thermally treated derivatives. a** TGA curves and the first derivative of TG for *p*TPTFA and *p*TPPFA. **b** TG-MS analysis of *p*TPPFA, highlighting the normalized intensity of selected ion signals released during thermal treatment. **c** Raman spectra of *p*TPPFA and thermally treated derivatives, with characteristic signals for carbon molecular sieve membranes marked with an asterisk (*). **d** Solid-state MAS ¹⁹F NMR (470 MHz) spectra of *p*TPPFA before and after thermal treatment, with spinning sidebands indicated by asterisks (*). **e** CP/MAS ¹³C NMR spectra (500 MHz) of *p*TPPFA and its thermally treated derivatives, showing changes in carbon bonding environments. **f** ATR-FTIR spectra of *p*TPPFA at different thermal treatment stages, illustrating structural transitions with defluorination. **g** Proposed structural transformations of *p*TPPFA upon thermal treatment (350–500 °C), emphasizing cross-linking mechanisms and radical formation.

(Supplementary Fig. 14) showed increased hardness and reduced modulus, indicating enhanced mechanical robustness typical of cross-linked polymers. In addition, tensile stress-strain measurements (Supplementary Fig. 15 and Table S2) confirm that *p*TPPFA membranes treated at 450 °C exhibit significantly improved tensile strength (68.9 ± 2.1 MPa) and Young's modulus (3.93 ± 0.46 GPa), accompanied by reduced elongation at break (2.39 ± 0.27%), illustrating a typical trade-off between stiffness and ductility[12,26]. As shown in Fig. 1c, the progressive color change from light to dark serves as a qualitative indicator of increasing aromaticity and densification, consistent with phenomena commonly reported in TR polymer systems[27]. While such transitions are often associated with increased rigidity and potential

brittleness, the *p*TPPFA 450 °C membranes retain flexibility, as demonstrated in Supplementary Movie 1 and 2, especially in contrast to CMS membranes carbonized at 675 °C. Moreover, these membranes maintain integrity under $CO_2/CH_4$ feed conditions at pressures as high as 850 psi. This performance further confirms their suitability for practical gas separation applications, which will be discussed in more detail later.

Spectroscopic analyses confirmed cross-linking via thermally induced defluorination. Solid-state magic-angle spinning (MAS) ¹⁹F NMR revealed a decrease in pentafluorobenzene peaks (−131 ppm ($F_b$, *o*), −153 ppm ($F_c$, *p*), −163 ppm ($F_d$, *m*)) with increasing temperature (400 °C to 500 °C, Fig. 2d). Overlapping peaks of $F_n$ at −125 ppm, −139

ppm, and −155 ppm were observed, which may correspond to fluor-obenzene and possibly some fluorine atoms in fluorinated biphenyl, resulting from treatment at 450 °C. Meanwhile, peaks corresponding to the trifluoromethyl group (−80– −50 ppm) persisted, indicating predominant defluorination of the pentafluorophenyl groups. This observation aligns with the concept that the dissociation energy of the carbon-fluorine bond increases with the number of fluorine atoms bonded to the same carbon; hence, the energy required to cleave the trifluoromethyl C-F bond exceeds that for the pentafluorobenzene C-F bond[28]. The thermally treated derivatives exhibited split tri-fluoromethyl peaks at −58 and −64 ppm. The peak at −58 ppm is attributed to changes in electron orientation of the carbon linked to fluorine, with the trifluoromethyl group located on a carbon linked to a pentafluorophenyl moiety at −64 ppm or a relatively less electron-withdrawing phenyl ring (−58 ppm)[29]. This may indicate the formation of new carbon-carbon bonds following defluorination by thermal activation. Furthermore, the solid-state cross-polarization/magic-angle spinning (CP/MAS) [13]C NMR spectra of thermally treated $p$TPPFA membranes revealed a predominantly aromatic structure. Peaks at 126 ppm and 139 ppm correspond to the terphenyl backbone (Fig. 2e). For $p$TPPFA 450 °C, a distinct peak at 115 ppm is assigned to newly formed aromatic C-C linkages, consistent with biphenyl-type structures. Although a weak signal is also observed at this chemical shift in the pristine $p$TPPFA membrane, its substantially increased intensity and sharper profile in the thermally treated derivatives strongly support its assignment to cross-linking-related structural evolution. Additionally, a small peak at 155 ppm is attributed to ortho-positioned $sp^2$ carbon atoms in the residual fluorinated biphenyl units. The increased inten-sity of the peak at 132 ppm further suggests a higher concentration of aromatic C-C linkages in the $p$TPPFA 450 °C compared to $p$TPPFA 400 °C, indicating progressive structural transformation toward a more densely cross-linked network.

As evidenced by the attenuated total internal reflection Fourier transform infrared (ATR-FTIR) spectra of thermally treated $p$TPPFA (Fig. 2f), partial cleavage of C-F bonds occurs during the thermally activated defluorination process, as indicated by the C-F stretching vibrations observed in the range of 1250 cm$^{-1}$ to 1100 cm$^{-1}$. Defluor-ination leads to changes in the aromatic ring system, evidenced by a shift of the aromatic C=C stretching vibration from 1480 cm$^{-1}$ to 1495 cm$^{-1}$, attributed to the defluorination of pentafluorophenyl groups. Additionally, a peak at 886 cm$^{-1}$, commonly associated with *meta*-substituted phenyl groups[30], suggests that some newly formed aromatic C-C linkages may involve *meta*-connected units. While para-substitution is generally more favorable, our results do not rule out the coexistence of both meta and para linkages in the thermally cross-linked network. We propose that the degradation of trifluoromethyl and pentafluorobenzene groups facilitates the formation of an arbi-trarily connected aromatic structure. Additionally, the F/C ratio cal-culated by the XPS survey spectra (Supplementary Fig. 16) decreased with increasing temperature, in agreement with [19]F NMR results. Spe-cifically, the F/C atomic ratio for $p$TPPFA treated at 450 °C reduced to 0.17, compared to the pristine $p$TPPFA ratio of 0.29, indicating the removal of the majority of fluorine atoms. Further evidence of inter-chain cross-linking is given by detailed scans of the C 1$s$ XPS spectra, which show a decrease in the intensity of peaks corresponding to the CF$_3$ or C-F groups, along with a shift in the peaks corresponding to C-C bonds, from 284.4 to 284.6 eV for $p$TPPFA and $p$TPTFA membranes treated at or above 450 °C. This supports the formation of new C-C linkages. Moreover, the thermally treated membranes could not be dissolved in N-methyl-2-pyrrolidone, indicating a structural change in the polymer at elevated heat treatment temperatures (Supplemen-tary Fig. 17).

Building on the characterization results (Fig. 2a–f), we propose the mechanism for the formation of EMPMs shown in Fig. 2g. Thermal treatment below the backbone decomposition temperature selectively activates fluorine atoms in pentafluorophenyl groups, generating reactive radicals. TG-MS analysis confirmed the release of CF$_3$, HF, and C$_6$F$_5$ fragments, and MAS [19]F NMR validated bond cleavage. These radicals drive inter-chain cross-linking (Fig. 2e), forming fluorinated biphenyl linkages and possibly increasing inter-chain distance[31]. Sub-sequent thermal decomposition of residual pentafluorophenyl groups generates microporosity. During the isothermal phase, polymer chain rearrangement stabilizes the cross-linked network, yielding a micro-porous structure with enhanced permeability and plasticization resistance, which will be discussed later.

## Structural and gas transport analysis

The thermally treated $p$TPPFA membranes exhibited a noticeable increase in average $d$-spacing, from 4.1 Å to 5.5 Å, with increasing thermal treatment temperature (Fig. 3a). These XRD results highlight that defluorination promotes a more open amorphous structure[22]. Additionally, N$_2$ adsorption/desorption isotherms at 77 K revealed that $p$TPTFA 500 °C attained a BET surface area of 158 m$^2$ g$^{-1}$, while $p$TPPFA 450 °C achieved an even higher value of 552 m$^2$ g$^{-1}$. In contrast, both pristine $p$TPTFA and $p$TPPFA membranes exhibited negligible N$_2$ uptake (Fig. 3b), further confirming the formation of a permanent microporous structure. According to the IUPAC classification[32], $p$TPPFA 500 °C exhibits a Type I isotherm, characteristic of CMS membranes[23]. However, $p$TPPFA 450 °C shows significant adsorption at low pressure (p/p$_0$ < 0.1) and displays continued N$_2$ adsorption with distinct hysteresis, which is consistent with the swelling behavior of polymer chains during N$_2$ adsorption. This hysteresis can be attributed to the absence of a fixed framework, similar to the behavior observed in PIMs[33]. The swelling and the corresponding hysteresis suggest that the polymer matrix may contain restricted-access pores or inter-connected free-volume elements that reorganize under gas sorption conditions, which is not typical of traditional solid sorbents such as zeolites or MOFs. This distinct sorption behavior reflects the non-equilibrium glassy nature of $p$TPPFA 450 °C, which provides a dynamic free volume for sorption but lacks the rigid structure to resist dilation under sorbate-induced stress.

Our hypothesis suggests that bonding between terphenyl blocks rigidifies the polymer chains, effectively "freezing" the free volume. A comparable outcome was observed in our recent work with a blend of carboxylated polyimide and ladder-structured amino-poly-silsesquioxane (LPSQ)[34]. In that study, amidation-induced thermal cross-linking between the carboxylic groups of polyimide and the amine groups of LPSQ produced larger and more interconnected cavities, as rigid double-stranded siloxane backbones disrupted chain packing. Similarly, the N$_2$ uptake of $p$TPPFA 450 °C is comparable to that of thermally rearranged polymers, although it remains slightly lower than that of PIM-1[27,35]. Notably, $p$TPTFA-containing phenyl groups also exhibited a modest increase in N$_2$ uptake after thermal treatment at 500 °C. Moreover, no D and G peaks were observed in the Raman spectra of $p$TPTFA 500 °C, consistent with the behavior of $p$TPPFA 450 °C (Supplementary Fig. 19). Consistent with the aforementioned results, the extrinsically microporous polymer $p$TPTFA 500 °C exhib-ited a smaller $d$-spacing compared to $p$TPPFA 450 °C, with values of 0.51 nm and 0.55 nm, respectively (Supplementary Fig. 20). This sug-gests that some trifluoromethyl groups degrade at elevated tempera-tures, leading to the formation of trityl radicals. These radicals are more likely to undergo dimerization, forming a Gomberg's dimer with a quinoid structure rather than a hexaphenylethane dimer (Supplemen-tary Fig. 21)[36]. Such cross-linked networks may lead to the formation of interconnected micropores similar to those observed in $p$TPPFA.

Consistent with these findings, the micropore size distribution, calculated from CO$_2$ adsorption isotherms at 273 K, exhibited a bimodal pore distribution, with pore volume increasing with tem-perature (Fig. 3c). The pore sizes of the thermally treated membranes (4–7 Å) were slightly smaller than those of PIM-1 (4–9 Å)[10].

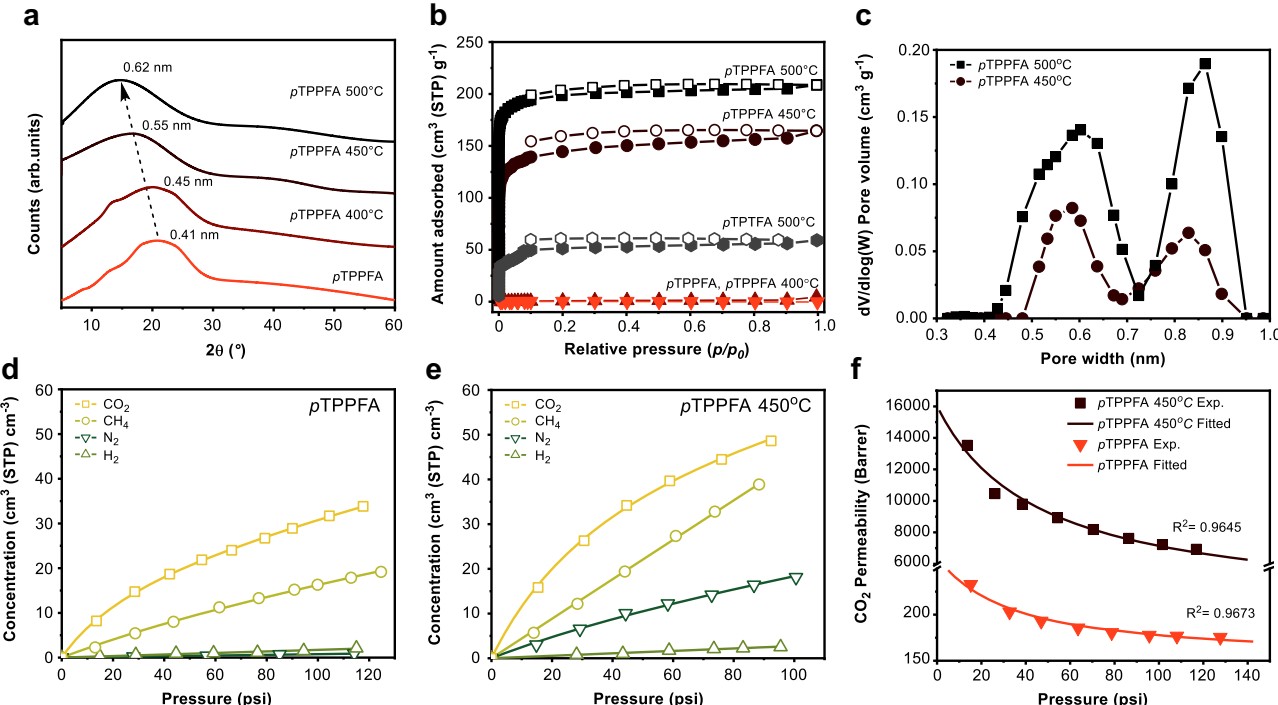

**Fig. 3 | Structural analysis of fluorinated aromatic polymer precursor and their thermally treated derivatives. a** WAXD patterns, **b** 77 K $N_2$ adsorption (filled) and desorption (open) isotherms of $p$TPPFA and thermally treated $p$TPPFA measured at 77 K. Textural properties of $p$TPPFA are summarized in Supplementary Table 3. **c** Pore size distribution of $p$TPPFA 450 °C and $p$TPPFA 500 °C calculated from $CO_2$ adsorption isotherms at 273 K (Supplementary Fig. 18). Gas adsorption isotherms of **d** $p$TPPFA, **e** $p$TPPFA 450 °C. **f** $CO_2$ permeability of $p$TPPFA and $p$TPPFA 450 °C as a function of pressure at 35 °C plotted in accordance with Eq. (7).

Additionally, gas adsorption experiments conducted at 35 °C showed a significant increase in gas uptake for $H_2$, $CO_2$, $N_2$, and $CH_4$ following thermal treatment, which can be attributed to the augmented surface area (Fig. 3d, e). The dual-mode sorption parameters, obtained through non-linear regression of the experimental data, are listed in Supplementary Table 4. Gas absorbed according to Henry's law behaves similarly to gas dissolving in a liquid or rubbery polymer, where solubility increases linearly with pressure. In contrast, gas absorbed via the Langmuir mode exhibits a finite capacity, corresponding to the filling of specific vacant sites. The affinity parameter $b$ and Henry's constant $k_D$ decreased due to the reduced fluorine content in $p$TPPFA 450 °C, as previously mentioned. However, the Langmuir capacity $C'_H$, increased significantly, owing to the formation of extrinsic micropores.

Analysis of the $CO_2$ transport results using the dual-mode transport model provides insight into changes in the membrane structure after defluorination[37]. The $CO_2$ permeability of $p$TPPFA and $p$TPPFA 450 °C membranes is shown in Fig. 3f as a function of pressure. According to the partial immobilization model, the permeation process of penetrant molecules in glassy polymeric membranes occurs within two distinct domains: the dense equilibrium structures of the polymers (dissolved mode) and the non-equilibrium excess volume (Langmuir mode). In this model, $D_D$ and $D_H$ represent the diffusivities in the dissolved and Langmuir modes, respectively. The parameters of the partial immobilization model were evaluated using a non-linear regression method for a plot of permeability ($P$) versus feed pressure ($p$) to Eq. (7), as listed in Supplementary Table 5. The increase in $D_D$ for $CO_2$ after cross-linking indicates looser polymer chain packing (5.74 ×$10^{-7}$ cm$^2$ s$^{-1}$ vs 1.58 ×$10^{-5}$ cm$^2$ s$^{-1}$), consistent with the WAXD results of the membranes. More prominent is the rise in $D_H$ for $CO_2$ (8.51 ×$10^{-8}$ cm$^2$ s$^{-1}$ vs 4.77 ×$10^{-6}$ cm$^2$ s$^{-1}$), which aligns with the formation of permanent interconnected microporosity, significantly facilitating gas diffusion compared to conventional glassy polymers. The substantial increase in the diffusion coefficient suggests that the microporous

structure provides additional pathways for gas molecules, resulting in significantly improved permeability.

## Temperature-Dependent Separation Performance of EMPMs

To comprehend the effect of defluorination-induced thermal cross-linking on gas transport, we evaluated the gas permeability of $H_2$, $CO_2$, $N_2$, and $CH_4$ in a series of $p$TPPFA and $p$TPPFA membranes treated at different temperatures under 1 bar and 35 °C (Fig. 4a). As expected, increasing the thermal treatment temperature enhanced gas permeability for all gases. Notably, the $p$TPPFA membrane treated at 450 °C exhibited a two-order-of-magnitude increase in $CO_2$ permeability compared to the pristine $p$TPPFA membrane (rising from 280 to 12,000 Barrer) while maintaining its glassy nature. This performance surpasses that of thermally rearranged (TR) polymer membranes (2000–4000 Barrer)[8] and is comparable to spirobisindane-based PIM[7] and Tröger's base PIM membranes[9]. In contrast, the $p$TPTFA 500 °C membrane displayed lower gas permeability for all gases, likely due to reduced micropore formation. This reduction can be attributed to differences in pendant groups, which influence the development of interconnected pores during thermal treatment. Nevertheless, the $p$TPTFA 500 °C membrane still exhibited significant permeability improvements compared to its untreated counterpart, demonstrating that thermal defluorination effectively transforms glassy polymeric membranes into microporous analogues.

The enhanced gas permeability of $p$TPPFA following defluorination can be attributed to significant increases in both diffusivity and solubility (Fig. 4b), with diffusivity playing a particularly prominent role in the permeability boost. For instance, $CO_2$ solubility increased by a factor of 3.9, while $CO_2$ diffusivity rose by a factor of 15.5. The $CO_2/N_2$ diffusivity selectivity showed a modest increase from 1.45 to 1.63, while solubility selectivity decreased from 16 to 8 (Supplementary Fig. 22). This pattern is consistent with previous studies, which suggest that the formation of new cavities tends to enhance the solubility of larger gas molecules more than that of smaller ones[38,39].

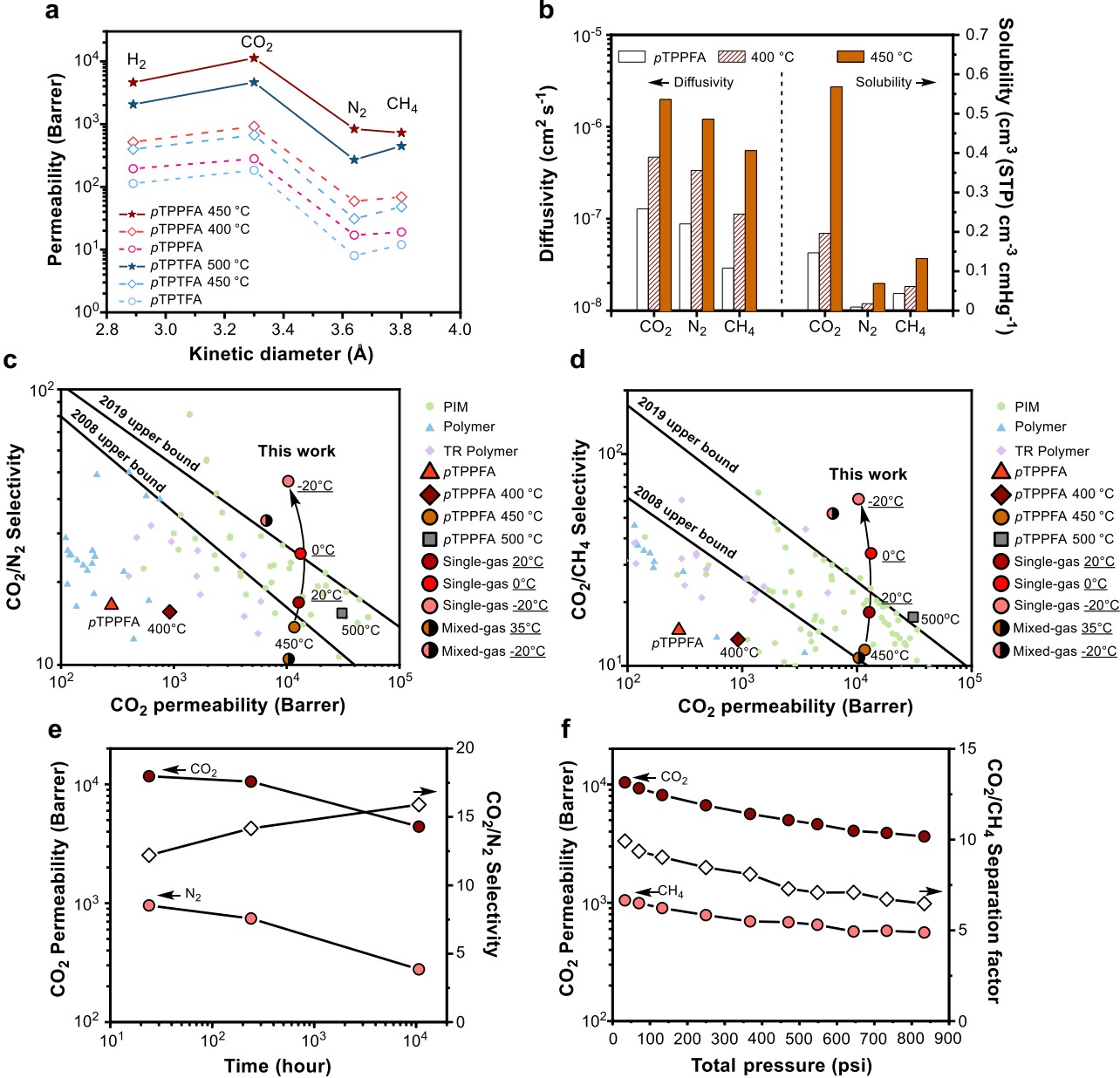

**Fig. 4 | Gas transport properties of EMPMs. a** Gas permeability of gases with different kinetic diameters in FAPs and their thermally treated derivatives at 35 °C and 1 bar. **b** Diffusivity and solubility of $CO_2$, $N_2$, and $CH_4$ for $p$TPPFA and its thermally treated derivatives. Comparisons of the gas separation performance of $p$TPPFA and thermally treated $p$TPPFA with state-of-the-art polymeric membranes for **c** $CO_2/N_2$, **d** $CO_2/CH_4$. The symbols used are as follows: triangles for $p$TPPFA, diamonds for thermally treated at 400 °C, circles for 450 °C, and squares for 500 °C. Black-edged circles indicate separation performances of $p$TPPFA 450 °C measured at −20 °C, 0 °C, 20 °C, and 35 °C. The underlined text indicates the measurement temperature. Note that filled symbols represent single-gas separation performance, while half-filled symbols represent mixed-gas separation performance. **e** Long-term $CO_2/N_2$ separation performance of $p$TPPFA thermally treated at 450 °C measured at 1 bar, 35 °C. **f** Mixed-gas $CO_2/CH_4$ separation performance of $p$TPPFA thermally treated at 450 °C as a function of total feed pressure at 35 °C.

The cold membrane process has gained attention as a promising approach for energy-efficient $CO_2$ capture, particularly for biogas upgrading and post-combustion gas separation[40,41]. For these processes to be viable, membranes must combine high intrinsic selectivity with sufficient permeability. As shown in Fig. 4c, d, the $p$TPPFA 450 °C membrane demonstrates impressive $CO_2$ separation performance, surpassing the 2019 upper bound for ultrapermeable benzotriptycene-based polymers[42], making it an attractive candidate for the cold membrane process. At 35 °C, $p$TPPFA 450 °C exhibits moderate $CO_2/N_2$ and $CO_2/CH_4$ selectivities of 13 and 15, respectively. However, lowering the temperature to −20 °C enhances $CO_2$ adsorption and restricts the diffusion of larger $N_2$ and $CH_4$ molecules, boosting selectivities to 46 and 61, respectively. This temperature-dependent improvement aligns

with findings by Ji et al. who observed significant improvements in PIM-1 performance at sub-ambient temperatures, with $CO_2/N_2$ selectivity rising from 21.1 at 35 °C to 55.6 at −20 °C, despite a 48.7% decrease in $CO_2$ permeability at −20 °C[43]. Notably, the $CO_2$ permeability of $p$TPPFA 450 °C remains relatively stable (10,324 Barrer) even at −20 °C, resulting in a remarkable increase in separation performance beyond the 2019 upper bound. This combination of high selectivity and permeability under sub-ambient conditions positions the EMPM as a promising candidate for energy-efficient $CO_2$ separation in industrial applications. Mixed-gas permeation results further confirmed the effective $CO_2$ separation performance of $p$TPPFA 450 °C at sub-ambient temperatures. Additionally, at 35 °C and 2 bar, the membrane exhibited increased permeabilities for highly condensable

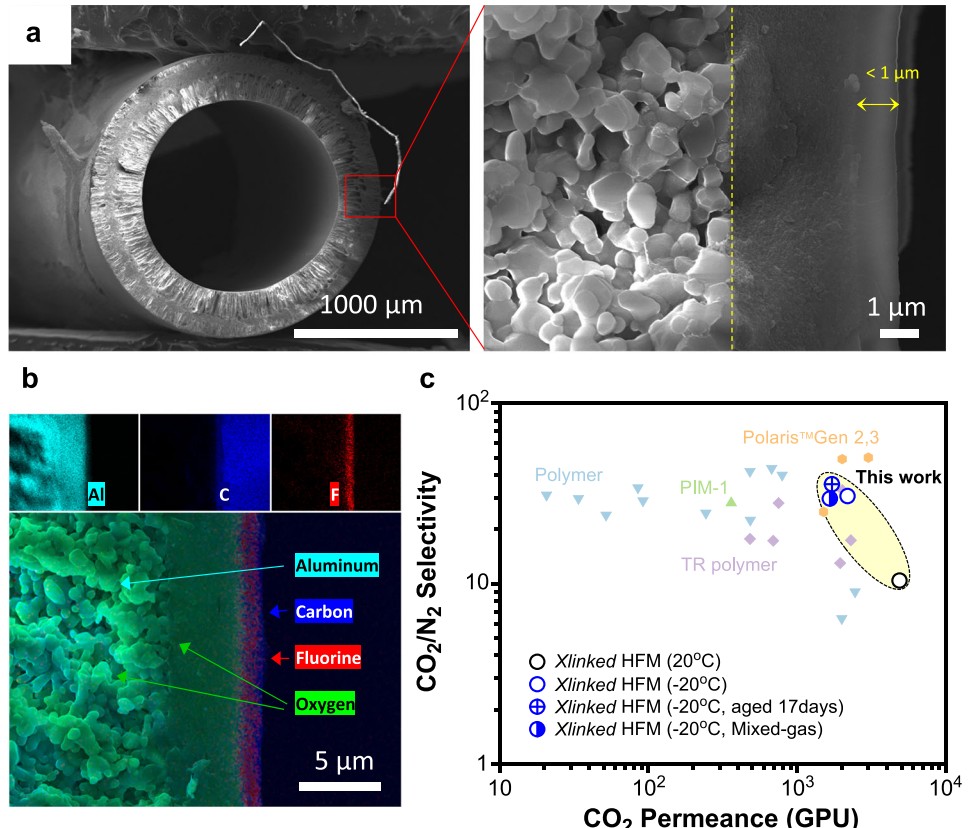

**Fig. 5 | Asymmetric *p*TPPFA EMP Hollow Fiber Membrane. a** Field emission scanning electron microscopy (FE-SEM) images. **b** Energy dispersive spectroscopy (EDS) analysis of the asymmetric *p*TPPFA EMP HFM. **c** Comparisons of $CO_2/N_2$ separation performance of *p*TPPFA 450 HFM (this work) with state-of-the-art membranes, including PIM-1 hollow fibers[51], TR hollow fibers[8,52], and Polaris™ (thin-film composite)[53]. Note that the cross-centered symbol represents the single-gas separation performance of 17-day-aged *p*TPPFA 450 HFMs measured at −20 °C and 1 bar, while the half-filled symbol represents the $CO_2/N_2$ mixed-gas separation performance of 18-day-aged *p*TPPFA 450 HFMs measured at −20 °C and 2 bar. Detailed data are provided in Supplementary Tables 6–8.

hydrocarbons ($C_2H_4$, $C_2H_6$, $C_3H_6$, and $C_3H_8$) along with moderate selectivities, suggesting the presence of micropores (<10 Å) in the thermally treated membrane (Supplementary Fig. 23).

**Long-term stability and plasticization resistance**
The *p*TPPFA 450 °C membrane demonstrates remarkable resistance to aging, maintaining a $CO_2$ permeability above 4000 Barrer even after 523 days (or 10,000 h) of vacuum storage (Fig. 4e). Physical aging, which typically reduces permeability in PIMs due to polymer chain densification and free volume loss, is substantially suppressed in this system. The aged *p*TPPFA 450 °C membrane still exhibits competitive $CO_2$ permeability compared to state-of-the-art membranes[44]. This long-term durability is attributed to the cross-linked network, which restricts chain mobility, mitigates aging, and preserves high $CO_2$ permeability over prolonged periods. In addition to its long-term stability under vacuum storage, the membrane also demonstrates robust operational durability during continuous mixed-gas conditions. A 96 h permeation test using a 50/50 mol% $CO_2/CH_4$ mixture at 35 °C and 2 bar resulted in only a 7.14% decrease in $CO_2$ permeability, underscoring its resistance to operational physical aging under realistic conditions (Supplementary Fig. 24). Lai *et al.* recently reported significant selectivity enhancement in aged hydrocarbon ladder polymers, achieved through three-dimensional backbone design and aging-induced narrowing of transport pathway, with minimal loss in permeability[45]. While the EMPM in this study exhibited only a modest increase in selectivity after aging, further improvements may be possible by introducing non-planar monomers to form rigid, three-dimensional polymer networks, potentially enabling aging-assisted

size–sieving enhancement. To assess plasticization resistance, $CO_2/CH_4$ mixed-gas permeation experiments were performed on the *p*TPPFA 450 °C membrane at 35 °C using an equimolar $CO_2/CH_4$ feed (Fig. 4f). The EMPM showed no signs of plasticization up to a feed pressure of approximately 40 bar, contrasting with PIM-1 membranes[46], which display plasticization-induced increases in $CO_2$ permeability above 8 bar, and the pristine *p*TPPFA membrane, which shows plasticization at around 10 bar (Supplementary Fig. 25). Both $CO_2$ and $CH_4$ permeabilities decreased with increasing feed pressure, consistent with dual-mode sorption effects[47]. These findings highlight that defluorination-induced thermal cross-linking of perfluorinated aromatic polymers presents a promising strategy for developing highly permeable, plasticization-resistant membranes. Future research may explore the incorporation of rigid, non-planar aromatic monomers in the polymer backbone to further disrupt chain packing more effectively and promote aging-assisted size-sieving, thereby enhancing gas separation performance.

**Fabrication of the EMP hollow fiber membrane**
Asymmetric *p*TPPFA hollow fiber membranes (HFMs) were prepared by sequentially dip-coating a γ-$Al_2O_3$ interlayer and a *p*TPPFA selective layer onto the outer surface of a porous α-$Al_2O_3$ hollow fiber substrate (Supplementary Note. 3). The *p*TPPFA HFM was then thermally treated at 450 °C for 2 h, is hereafter referred to as *p*TPPFA 450 HFM. Cross-sectional SEM images and EDS analysis confirmed that the thermally treated *p*TPPFA polymer chains tightly interlocked with the γ-$Al_2O_3$ interlayer, with no signs of delamination between the gutter and selective layers (Fig. 5a, b). The selective layer was approximately 1 μm thick.

The $p$TPPFA 450 HFM achieved an impressive $CO_2$ permeance of 4848 GPU with a moderate $CO_2/N_2$ selectivity of 10.4 at 20 °C, attributed to its microporous structure (Fig. 5c). For comparison, Liu et al. reported a Matrimid® polyimide HFM with a $CO_2$ permeance of 143 GPU and a $CO_2/N_2$ separation factor of 80.3 at −20 °C for sub-ambient applications[48]. Our $p$TPPFA 450 HFM exhibited a substantially higher $CO_2$ permeance of 2174 GPU and a $CO_2/N_2$ selectivity of 30.6 at −20 °C. After 17 days of aging, it maintained a high $CO_2$ permeance (1708 GPU), with a slight increase in $CO_2/N_2$ selectivity to 35.6. Mixed-gas testing showed a $CO_2$ permeance of 1654 GPU and a $CO_2/N_2$ separation factor of 29.6, with minor decreases due to the competitive sorption effect between $CO_2$ and $N_2$. The high permeance of the $p$TPPFA 450 HFM suggests it could lower capital costs in membrane-based $CO_2$ capture systems, making it a strong candidate for sub-ambient $CO_2$ separation applications.

## Discussion

In summary, we report a class of microporous polymer membranes with high resistance to plasticization, termed EMPM, developed through defluorination of aromatic glassy polymers. In particular, the reorganization of terphenyl blocks during thermal treatment of per-fluorinated, aryl-ether-free polymers facilitates the formation of a permanent microporous structure by disrupting polymer chain packing. The resulting EMPM exhibits a BET surface area of 552 $m^2 g^{-1}$ and a pore volume of 0.23 $cm^3 g^{-1}$. Notably, it demonstrates effective gas separation performance for various gas pairs, including $CO_2/N_2$, $CO_2/CH_4$, $C_2H_4/C_2H_6$, and $C_3H_6/C_3H_8$, surpassing the respective polymeric upper bound limits. Additionally, the EMPM enhances resistance to plasticization and long-term stability. The gas transport behavior is interpreted using a dual-mode transport model, providing mechanistic insights into the separation characteristics of microporous polymers. Furthermore, the asymmetric, extrinsically microporous hollow fiber membrane demonstrates effective $CO_2/N_2$ separation performance, confirming its potential for industrial application. This thermally induced defluorination strategy provides a practical route to microporous polymeric materials derived from fluorinated aromatic glassy polymers, supporting high-throughput fabrication of hollow fiber membranes for advanced gas separation processes.

## Methods

### Materials

$Para$-terphenyl (99.5%), 2,2,2-trifluoroacetophenone (99%), and trifluoromethanesulfonic acid (≥99%), dichloromethane (DCM, ≥99.8%) were acquired from Sigma-Aldrich. Octafluoroacetophenone (97%) was obtained from Apollo Scientific. Methanol (99.5%) and tetrahydrofuran (THF, 99.8%) were purchased from Daejung Chemical Co., LTD. $CO_2$, $N_2$, $CH_4$, $C_2H_4$, $C_2H_6$, $C_3H_6$, and $C_3H_8$ (all >99.5%) were acquired from Shin Yang Oxygen Co., LTD.

### Synthesis of Fluorinated Aromatic Polymers

The $p$TPTFA and $p$TPPFA polymers were synthesized via superacid-catalyzed polycondensation of fluorinated acetophenone with aromatic hydrocarbons[49]. For the synthesis of $p$TPTFA, 2,2,2-trifluoroacetophenone (2.61 g, 15.0 mmol), $para$-terphenyl (3.45 g, 15.0 mmol), trifluoromethanesulfonic acid (16 mL), and DCM (14 mL) were added to a 100 mL three-neck round-bottom flask and reacted at 4 °C for 72 h, producing a viscous dark blue precipitate. This precipitate was washed with methanol and then extracted with hot methanol, yielding 5.82 g. For the synthesis of $p$TPPFA, octafluoroacetophenone (3.96 g, 15.0 mmol), $para$-terphenyl (3.45 g, 15.0 mmol), and trifluoromethanesulfonic acid (1.3 mL) were added to a 10 mL three-neck round-bottom flask and reacted at room temperature for 16 h, resulting in a transparent, viscous orange solution. This solution was slowly poured into water to form white fibers, which were washed with water and extracted with hot methanol, yielding 7.03 g.

### Preparation of $p$TPPFA and $p$TPTFA membranes and EMPMs

The $p$TPTFA and $p$TPPFA powders were dissolved in THF at a concentration of 4 wt% to prepare the dope solutions. These solutions were cast onto a clean glass plate using a Teflon ring mold within a THF-saturated glove bag at room temperature. After 48 h, the vitrified films, with a thickness of 80 ± 10 μm, were dried at 120 °C for 24 h under vacuum to remove any residual solvent. The $p$TPPFA films then underwent thermal treatment at specific final temperatures ($T_{final}$) of 400 °C, 450 °C, or 500 °C in a three-zone furnace (Thermcraft). The process involved heating the membranes from 50 to 250 °C at 13.3 °C $min^{-1}$, followed by further heating to $T_{final}$ °C at 3.85 °C $min^{-1}$. The membranes were then soaked at $T_{final}$ for 2 h, after which the furnace was naturally cooled down under an argon (99.999%) purge. After each run, the quartz tube and quartz plate were rinsed with acetone and purged with air at 800 °C.

### Supplementary Characterizations

To detect evolving gases during thermal treatment, a simultaneous thermal analyzer (STA 409 PC, Netzsch) coupled with Netzsch QMS 403 C mass spectroscopy was used, with a 3 °C $min^{-1}$ ramp rate under an argon environment. Raman spectra of the membranes were obtained using an InVia Raman microscope (Renishaw) equipped with a Nd:Yag laser (532 nm), scanning from 4000 $cm^{-1}$ to 200 $cm^{-1}$ with a 2 $cm^{-1}$ beam resolution. Solid-state CP/MAS $^{13}C$, MAS $^{19}F$ NMR spectra were acquired using a Bruker Advance III HD 500 MHz spectrometer equipped with a 2.5 mm solid-state CP/MAS probe under ambient air conditions at 25 °C. The spinning frequency for $^{13}C$ was set at 15 kHz, while for $^{19}F$, it was varied between 10 kHz and 15 kHz to investigate the dependence of spin exchange on MAS frequency. The $^{19}F$ chemical shifts were referenced to the −204 ppm chemical shift of LiF. ATR-FTIR spectroscopy was conducted using a Nicolet 380 (Thermo Scientific) within the range of 4000 $cm^{-1}$ to 650 $cm^{-1}$, with a resolution of 4 $cm^{-1}$. Wide-angle x-ray diffraction (WAXD) patterns were obtained using a D/MAX-2500H (Rigaku) instrument with CuKα radiation, spanning from 5° to 40° with a scan rate of 2° $min^{-1}$ at 40 kV and 100 mA. $N_2$ adsorption isotherms at 77 K and $CO_2$ adsorption isotherms at 273 K were used to determine the surface area, pore volume, and pore size distribution of thermally treated membranes. The isotherms were measured using ASAP 2020 Plus (Micromeritics) following pretreatment at 120 °C under vacuum for 24 h. Non-local density functional theory (NLDFT) was applied to compute the pore size distribution of thermally treated membranes based on $CO_2$ adsorption at 273 K.

### Gas adsorption characterizations

Adsorption isotherms for $H_2$, $CO_2$, $N_2$, and $CH_4$ were measured in the pressure range of 14 to 130 psi at 35 °C using a dual-volume sorption cell and the pressure decay method[50].

### Single-gas and mixed-gas permeation

Single-gas permeation experiments were carried out for $CO_2$, $N_2$, $CH_4$, $C_2H_4$, $C_2H_6$, $C_3H_6$, and $C_3H_8$ using the constant volume/variable pressure method at 35 °C. A detailed description of the sub-ambient experiments can be found in Supplementary Note 5 and Supplementary Fig. 26.

Permeability was calculated using the following Eq. (1).

$$P = \frac{22,414 \cdot V \cdot l}{R \cdot T \cdot \Delta p \cdot A} \times \frac{dp}{dt} \tag{1}$$

Where $P$ represents gas permeability in Barrer units (1 Barrer = $10^{-10}$ $cm^3$ (STP) cm $cm^{-2}$ $s^{-1}$ $cmHg^{-1}$), $V$ is the downstream volume, $l$ denotes the membrane thickness, $R$ is the gas constant, $T$ indicates temperature, $\Delta p$ is the pressure difference across the membrane, $A$ is the effective membrane area, and $dp/dt$ is the downstream pressure change over time under steady-state conditions.

The average solubility $S$, in cm$^3$ (STP) cm$^{-3}$ cmHg$^{-1}$, was calculated from gas adsorption isotherm results using:

$$S = \frac{C}{p} \qquad (2)$$

Where $C$ is the concentration of gas in the membrane, and $p$ is the feed pressure.

The average diffusivity $D$, in cm$^2$ s$^{-1}$, was calculated by dividing permeability by solubility:

$$D = \frac{P}{S} \qquad (3)$$

Mixed-gas permeation tests for $CO_2/N_2$ and $CO_2/CH_4$ (50/50 mol/mol) were conducted using the constant volume/variable pressure method at 35 °C. To avoid concentration polarization, the stage cut was kept below 1%. Gas chromatography (GC, 7890 A (Agilent)) with a thermal conductivity detector was employed to analyze the downstream gas composition. Mixed-gas permeability was determined using:

$$P_i = \frac{22,414 \cdot V \cdot l}{R \cdot T \cdot \Delta p_i \cdot A} \times \frac{dp_{total}}{dt} \times y_i \qquad (4)$$

Where $\Delta p_i$ represents the partial pressure difference of component $i$ across the membrane, $dp_{total}/dt$ indicates the steady-state total pressure increase downstream, and $y_i$ denotes the mole fraction of component $i$, determined from GC area readings. The separation factor ($\alpha^*_{i/j}$) was calculated as follows:

$$\alpha^*_{i/j} = \frac{\left(y_i/y_j\right)}{\left(x_i/x_j\right)} \qquad (5)$$

where $x$ and $y$ are the upstream and downstream mole fractions, respectively.

## Dual-mode transport model

The one-dimensional flux of gas $A$ permeating through a glassy polymer is described by the dual-mode model as:

$$N_A = D_D \frac{\partial C_D}{\partial x} - D_H \frac{\partial C_H}{\partial x} \qquad (6)$$

where $D_D$ and $D_H$ represent the diffusion coefficients in the Henry's law and Langmuir domains, respectively. $C_D$ is the penetrant concentration in the Henry's law domain, and $C_H$ is the penetrant concentration in the Langmuir domain.

The pressure-dependent permeability $P$ of a glassy polymer membrane is given by:

$$P = k_D D_D + \frac{C'_H b D_H}{1 + bp} \qquad (7)$$

$k_D$ represents the Henry's law constant (cm$^3$ (STP) cm$^{-3}$ atm$^{-1}$), $C'_H$ stands for the Langmuir capacity constant (cm$^3$ (STP) cm$^{-3}$), and $b$ denotes the Langmuir affinity parameter (atm$^{-1}$). The parameters $C'_H$, $b$, and $k_D$ were determined by fitting the $CO_2$ adsorption isotherms of $p$TPPFA and $p$TPPFA 450 °C. $D_D$ and $D_H$ (cm$^2$ s$^{-1}$) were calculated by fitting the $CO_2$ permeability data obtained at various feed pressures to Eq. (7).

## Data availability

All data supporting the findings of this study are available within the article and the Supplementary Information, or available from the corresponding author upon request. Source data are provided with this paper.

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

## Acknowledgements

J.H.S. and J.S.L. acknowledge support from the National Research Foundation of Korea (NRF) under grants RS-2024-00356376 and 2020R1A5A1019631, and from the Korea Institute of Energy Technology Evaluation and Planning (KETEP) grant funded by the Korea government (MOTIE) (20212010200110, Development of compact $CO_2$ capture technology for combustion exhaust gas of urban LNG power plants). A.S.L. acknowledges funding from the National Research Council of Science & Technology (NST) grant funded by the Korea government (MSIT) (CRC22031-000). We acknowledge the assistance of Dr. Iqubal Hossain for his valuable comments on the structural analysis of the cross-linked membranes, and Dr. Jin Woo Oh for his contributions to the design and construction of the in-house permeation system (IGLOO) for sub-ambient temperature operation.

## Author contributions

J.H.S. and J.S.L. conceived the approach and designed the research. J.H.S. performed all the experiments, undertook data analysis and wrote the manuscript. H.J.Y. prepared the initial draft of the manuscript. H.A. provided constructive suggestions for results and discussion. J.J. and A.S.L. synthesized and characterized the polymers. J.H.P. provided the α-alumina hollow fiber substrate. J.S.L. supervised the project. All authors contributed to the discussion of the results and commented on the manuscript.

## Competing interests

J.H.S. and J.S.L. are inventors on US and Korean patent applications related to the membranes described in this work. The authors declare no other competing interests.
