## [Transparent Peer Review file · Nature Communications]

Extrinsically Microporous Polymer Membranes Derived from Thermally Cross-linked Perfluorinated Aryl-Ether Free Polymers for Gas Separation

Corresponding Author: Professor Jong Suk Lee

Version 0:

Reviewer comments:

Reviewer #1

(Remarks to the Author)
[Editor Note: See attached File]

Reviewer #2

(Remarks to the Author)

This paper reports crosslinked fluorinated polymer membranes with intrinsic porosity with high selectivity and high permeability, exceeding the latest upper bound. The overall materials design and experiments are nicely-conducted with reasonable analysis and literature survey. The work is useful for the separations and membrane community. However, the reviewer found two missing information. First is the justification of the use of fluorinated polymer membranes for membrane-based separations. Because of the extremely negative environmental impact of these forever chemicals (fluorinated compounds), the use of fluorinated polymers has been banned/restricted in the US and many developed nations, and there have been active research to isolate and destroy these fluorinated compounds. The motivation to specifically use fluorinated polymers for separations (although the recent restrictions) need to be included in the introduction. Second, the common issue with thermally crosslinked membrane is mechanical stability. A very brief few sentences were included, but not sufficient. Based on TR literature and color change (Fig 1), mechanical stabilities and film weakness should be further discussed.

The reviewer suggests the publication of the manuscript with revision.

A minor comment is, to be consistent with chemical structure nomenclature (especially with subscript, e.g., CH₄, O₂, etc.)

Reviewer #3

(Remarks to the Author)

In this work, authors synthesized a microporous membrane with cross-linked structure through high-temperature defluorination, which exhibits gas separation performance at low temperatures. However, the results are not solid to support the hypothesis or conclusion. The following points need to be considered.

1. The accompanying video can only show the soft behavior of obtain material. How about other mechanical properties (for example, toughness, strength)? Authors should add more videos to support other important mechanical properties.
2. Structure analyses should be carefully conducted. For example, authors mentioned that "the peak at 115 ppm is due to the newly formed aromatic C-C linkages in the biphenyl moiety" (Fig. 2e). However, the similar peak can also be found in the non-heat-treated membrane (pTPPFA).
3. The authors mentioned that "terphenyl block bonding primarily occurs at the meta position." (Fig.2g), but the reviewer did not find supporting evidence in the literature (ref. 28) and believes that it is more likely to occur in para position.
4. To comprehensively evaluate the membrane's durability and aging resistance, it is necessary to include graphs obtained under long-term continuous operating conditions. For example, a CO₂ permeability-time diagram spanning 72 hours of uninterrupted operation could serve as an indicator of the membrane's stability and performance consistency.

5. This material could work at low temperature and high pressures. However, this combination could be a drawback in practical applications due to the low efficiency and high requirements of operations. Authors should add more discussions to support the interest of this contribution.

6. It is well-known that C-F bond is among the strongest covalent bond. Authors hypothesized that the formation of C-C bonds took place by breaking the C-F bonds. It is a wired design based on the energetic reason. Authors should discuss literatures and design related tests to support the structure change.

7. Fig. 2b TG-MS test results show that the C6F5 signal appears at a temperature close to or even lower than that of the F signal, suggesting that C6F5 may be more easily removed than F, which is contrary to the authors' idea.

8. The sizes of tested gaseous molecules are at nanometer scale. But the SEM and related test are at micrometer scale, thousand times larger than the guest molecules. It is necessary to add more characterizations at a relevant scale to support this work.

Version 1:

Reviewer comments:

Reviewer #1

(Remarks to the Author)

The work summarizes the development of cross-linked membranes modified extrinsically microporous polymer membranes (EMPMs) as a novel class of microporous membranes, fabricated from perfluorinated aryl-ether-free aromatic polymers via defluorination-induced thermal cross-linking. Interesting trends in the gas selectivity, the separation performance is above the current upper bound lines for single and mixed-gas pairs. All our questions were answered one by one in the response, I agree to publish it now.

[Editor Note: This reviewer was asked to evaluate the rebuttal to Reviewer #2's comments, and provided the following:]

the authors has revised the manuscript carefully according to the reviewer's comments and could be accepted for publishing in NC. All of the technical concerns have been addressed now.

Reviewer #3

(Remarks to the Author)

The authors have properly addressed comments.

A detailed response to Reviewer's comments

We sincerely appreciate the editor and reviewers for their thoughtful and constructive feedback, which has been invaluable in improving the quality of our manuscript (NCOMMS-25-00242A). We have carefully addressed all the reviewers' comments and revised the manuscript accordingly. In addition to implementing the suggested changes, we have also incorporated the editorial recommendations. We hope that the revised manuscript meets the expectations of the reviewers and editor, and we respectfully submit it for consideration for publication in *Nature Communications*.

Reply to the comments by Reviewer # 1

This work summarizes the development of cross-linked membranes modified extrinsically microporous polymer membranes (EMPMs) as a novel class of microporous membranes, fabricated from perfluorinated aryl-ether-free aromatic polymers via defluorination-induced thermal cross-linking. Interesting trends in the gas selectivity, the separation performance is above the current upper bound lines for single and mixed-gas pairs. While the scope of characterization is comprehensive, there are some questions that need to be addressed to understand the underlying mechanism of what is occurring in these membranes.

Response: We are grateful to the reviewer for their thorough evaluation of our manuscript and for carefully highlighting the mechanism of defluorination-induced thermal cross-linking. Their valuable feedback has greatly contributed to improving the quality of our work.

Comment

1. According to the results of the characterization in the paper, the mechanism of *p*TPPFA membrane is carried out according to the route given in the paper when soaked in 450 °C, what will be the result when the soaked time is prolonged at a low temperature, such as 400 °C or 380 °C for 5-10 h, which will show the same results or the formation of CMS membrane.

Response: We sincerely appreciate the reviewer's insightful query. To address it, we prepared *p*TPPFA membranes thermally treated at 380 °C and 400 °C for extended soaking times of 10 hours. As shown in the Raman spectra (**Figure R1**), neither sample exhibited the characteristic D or G bands associated with carbon molecular sieve (CMS) structures, indicating that carbonization did not occur under these conditions. Additionally, 77 K N₂ adsorption measurements for the sample treated at 400 °C showed negligible uptake, suggesting the absence of microporosity. Notably, the modulated DSC plot (**Figure R2**) reveals an exothermic peak at 440 °C, even for samples pretreated at 400 °C for 2 hours, implying that microporosity formation is only initiated at higher temperatures. These results collectively suggest that prolonged thermal treatment below the threshold temperature associated with the decomposition of bulky fluorinated pendent groups is not sufficient to induce microporosity or CMS membrane formation.

Figure R1. Raman spectra of *p*TPPFA membranes after thermal treatment at various conditions. Characteristic D and G bands indicative of carbonization are observed only after treatment at 675 °C for 2 hours, whereas samples treated at 380 °C and 400 °C for 10 hours retain polymer features without signs of carbon structure formation.

Figure R2. Modulated DSC thermogram of *p*TPPFA after thermal treatment at 400 °C. The presence of an exothermic peak around 440 °C suggests that significant structural transitions, such as micropore formation, occur only at elevated temperatures beyond this point.

Comment

2. The author points out the nano-indentation results (Supplementary Fig. 14) showed increased hardness and reduced modulus, indicating enhanced mechanical robustness typical of cross-linked polymers, what about mechanical property (strain-stress).

Response: We appreciate the reviewer's constructive comment. To further assess the mechanical performance of the membranes, we have added stress-strain curves and a summary of the corresponding mechanical properties for both pristine *p*TPPFA and thermally treated *p*TPPFA 450 °C membranes in the Supplementary Information (**Figure R3** and **Table R1**). The results indicate that *p*TPPFA 450 °C exhibits improved mechanical robustness compared to the pristine polymer. Specifically, the thermally treated membrane shows higher tensile strength (68.9 ± 2.1 MPa vs. 60.2 ± 4.7 MPa) and a significantly increased Young's modulus (3.93 ± 0.46 GPa vs. 1.09 ± 0.07 GPa), indicating the formation of a stiffer and more rigid network due to thermal cross-linking. Conversely, the elongation at break decreases (2.39 ± 0.27 % vs. 8.22 ± 0.62 %), reflecting the expected reduction in ductility commonly observed in cross-linked systems. These results are consistent with the nano-indentation measurements (**Table R2** and **Supplementary Fig. 14**) and further corroborate the development of a mechanically reinforced, cross-linked structure in the thermally treated membrane. The relevant sentences have been incorporated into the revised manuscript.

Figure R3. Stress-strain curves of *p*TPPFA and *p*TPPFA 450 °C.

Table R1. Summary of mechanical properties of *p*TPPFA and *p*TPPFA 450 °C membranes obtained from tensile stress-strain measurements.

Properties	Tensile strength (MPa)	Elongation strain at break (%)	Young's modulus (Gpa)
p TPPFA	60.2 ± 4.7	8.22 ± 0.62	1.09 ± 0.07
p TPPFA 450 °C (EMPM)	68.9 ± 2.1	2.39 ± 0.27	3.93 ± 0.46

Table R2. Summary of mechanical properties of *p*TPPFA and *p*TPPFA 450 °C membranes obtained from nano-indentation measurements.

Properties	Hardness (GPa)	Reduced modulus (GPa)	Mean Contact Depth (nm)
p TPPFA	0.217 ± 0.002	3.15 ± 0.013	913 ± 3.34
p TPPFA 450 °C (EMPM)	0.277 ± 0.001	2.75 ± 0.010	507 ± 0.678

In the revised manuscript Page 9, Line 197-214:

“The thermally treated membranes (thickness: 60 ± 5 μm) retained their structural integrity. Nano-indentation analysis (Supplementary Fig. 14) revealed increased hardness and reduced modulus, indicating enhanced mechanical robustness consistent with cross-linked polymer behavior. In addition, tensile stress-strain measurements (Supplementary Fig. 15 and Table S2) confirm that *p*TPPFA membranes treated at 450 °C exhibit significantly improved tensile strength (68.9 ± 2.1 MPa) and Young's modulus (3.93 ± 0.46 GPa), accompanied by reduced elongation at break (2.39 ± 0.27 %), illustrating a typical trade-off between stiffness and ductility^{12,24}. As shown in Fig. 1c, the progressive color change from light to dark serves as a qualitative indicator of increasing aromaticity and densification, consistent with phenomena commonly reported in TR polymer systems²⁵. While such transitions are often associated with increased rigidity and potential brittleness, the *p*TPPFA 450 °C membranes retain excellent flexibility, as demonstrated in Supplementary Videos 1 and 2, especially in contrast to CMS membranes carbonized at 675 °C. Moreover, the membranes maintain integrity under CO₂/CH₄ feed conditions at pressures as high as 850 psi, further confirming their suitability for practical gas separation applications, which will be discussed in more detail later.”

In the revised Supplementary Note 1:

“Tensile tests were carried out using a Shimadzu Autograph AGS-X precision universal testing machine equipped with a 500 N load cell. Membranes with a thickness of 30-50 μm were cut into regular specimens with an effective length of approximately 20 mm and a width of approximately 8 mm. Young’s modulus was determined from the initial slope of the stress-strain curve. Tensile strength at break and elongation at break were obtained by averaging the results from three independent tests performed on separate membrane specimens.”

Comment

3. To explain the mechanism more clearly, is it possible to use TG-MS or TG-IR.

Response: We thank the reviewer for this constructive suggestion. In our study, we focused on elucidating the defluorination-induced cross-linking mechanism by integrating thermal analysis, spectroscopy, and porosity measurements.

Our proposed mechanism proceeds through multiple sequential steps: first, thermal cleavage of fluorinated pendant groups initiates defluorination. This leads to the generation of reactive aryl radicals, which undergo inter-chain cross-linking. This cross-linking increases the rigidity of the polymer matrix, thereby restricting chain mobility and setting the stage for structural reorganization.

To confirm the initial defluorination step, we performed thermogravimetric analysis coupled with mass spectrometry (TG-MS), as shown in **Figure 2b**. Upon heating above 400 °C, we observed the release of fluorine-containing fragments including F⁻ (m/z = 19), HCF₂⁺ (m/z = 51), CF₃⁺ (m/z = 69), and C₆F₅⁺ (m/z = 168), which confirms the detachment of pendant fluorinated groups and supports the onset of chemical activation. These fragments are attributed to side chain cleavage, as the main *para*-terphenyl backbone remains thermally stable up to 450 °C, as shown by TG-MS and GC-MS results.

To examine the chemical structure of the thermally treated polymer, we employed solid-state ¹⁹F and ¹³C NMR spectroscopy along with FT-IR analysis. The ¹⁹F MAS NMR spectra showed a significant decrease in signals corresponding to the pentafluorophenyl groups, indicating selective defluorination from pendant side chains. Meanwhile, the ¹³C CP-MAS NMR spectra revealed an enhanced peak, which we attribute to newly formed aromatic C–C linkages, consistent with radical coupling between polymer chains. Complementary FT-IR analysis supported these findings, showing the disappearance of characteristic C–F stretching bands and the appearance of new aromatic ring vibrations. Together, these spectroscopic results confirm that thermally induced defluorination leads to the formation of rigid aryl–aryl cross-links, contributing to the development of a stabilized polymer network.

At elevated temperatures, the constrained polymer chains undergo partial rearrangement, and the decomposition of remaining fluorinated moieties generates additional free volume. These combined effects culminate in the formation of extrinsic microporosity stabilized within

the rigid cross-linked network.

To examine the physical consequences of these transformations, we conducted wide-angle X-ray diffraction (WAXD) and nitrogen physisorption at 77 K. The d-spacing increased from 4.1 Å to 5.5 Å, and BET surface area increased to 552 m² g⁻¹ after thermal treatment at 450 °C. These results confirm the formation of micropores, reflecting the emergence of a rigid cross-linked architecture.

In summary, our integrated analysis of the thermal behavior, spectroscopic features, and structural evolution provides compelling evidence for a stepwise mechanism. We demonstrate that thermally induced defluorination initiates inter-chain cross-linking and that subsequent thermal reorganization stabilizes a microporous network within the polymer matrix.

Comment

4. Is *p*TPPFA can be made into Asymmetric *p*TPPFA hollow fiber membranes (HFMs), and *p*TPTFA can not?

Response: We thank the reviewer for this insightful question. Both *p*TPTFA and *p*TPPFA are solution-processable polymers and are amenable to fabrication into asymmetric hollow fiber membranes (HFMs) using dip-coating, a versatile method for forming thin, dense selective layers on porous supports. In our preliminary studies, both polymers demonstrated excellent solubility and film-forming properties. In this study, we focused on *p*TPPFA for hollow fiber membrane fabrication due to its superior gas separation performance following thermal treatment. Specifically, *p*TPPFA treated at 450 °C exhibited significantly higher CO₂ permeability than *p*TPTFA treated at 500 °C (12,162 vs. 4,632 Barrer). Although *p*TPTFA-based HFMs were not explored in this work, our findings indicate that both polymers are suitable candidates for asymmetric membrane fabrication.

Comment

5. Do author change the current structure, if put F or CF₃ in the triphenyl will have the current mechanism?

Response: We thank the reviewer for the insightful suggestion regarding the potential effect of introducing F or CF₃ substituents into the terphenyl backbone on the microporous membrane formation mechanism. We hypothesize that the current mechanism is driven by thermal defluorination, which generates aryl radicals that subsequently form new sp²-sp² carbon linkages. If fluorine or trifluoromethyl groups are incorporated into the terphenyl core, two plausible pathways may be considered. The first involves direct coupling between fluorinated terphenyl main chains via aryl radical recombination. However, due to the steric constraints associated with short sp²-sp² bonds and limited flexibility of the rigid backbone, this pathway is likely to contribute minimally. The second, and more probable, mechanism involves cross-linking between the fluorinated terphenyl backbone and the pentafluorophenyl side chains. In this case, defluorination-induced aryl radical formation could facilitate inter-chain linkages, leading to a rigid, cross-linked network. This would restrict polymer chain mobility and may promote microporosity. However, such materials may exhibit a lower BET surface area compared to EMPMs derived from *p*TPPFA treated at 450 °C, likely due to the absence of highly porous biphenyl-type linkages. Moreover, we would like to point out that the above cases are currently very difficult to prove experimentally at this time as there are no commercially available partially fluorinated terphenyl monomers that are able to be polymerized into aryl-ether-free polyaromatic polymers. We acknowledge that this would be an interesting work which would require monomer synthesis for future studies.

We sincerely appreciate the reviewer's comment, which raises an important structural design consideration. Exploring such backbone modifications could provide valuable insights into structure–property relationships and guide the rational design of future EMPM materials.

Case 1. Defluorination-induced cross-linking between fluorinated terphenyl blocks

Case 2. Defluorination-induced cross-linking between fluorinated terphenyl block and pentafluorophenyl group

Figure R4. Proposed reaction pathways for defluorination-induced cross-linking in fluorinated terphenyl-based polymers. Case 1 illustrates direct recombination between fluorinated backbones, while Case 2 involves radical coupling between the terphenyl core and pentafluorophenyl side groups upon thermal defluorination.

Comment

6. There are some typographical errors that need to be corrected, such as the lower corners or Spaces (400 oC), and so on.

Response: We thank the reviewer for pointing out the typographical errors. We have thoroughly reviewed the manuscript and corrected inconsistencies, including spacing issues (e.g., “400 oC” → 400 °C), use of subscripts/superscripts, and formatting of units and symbols to ensure consistency and clarity throughout the text.

Reply to the comments by Reviewer # 2

This paper reports crosslinked fluorinated polymer membranes with intrinsic porosity with high selectivity and high permeability, exceeding the latest upper bound. The overall materials design and experiments are nicely-conducted with reasonable analysis and literature survey. The work is useful for the separations and membrane community. However, the reviewer found two missing information... The reviewer suggests the publication of the manuscript with revision...

Response: We sincerely appreciate the reviewer's positive and constructive feedback. We are pleased that the reviewer found the materials design, experimental work, and analysis to be well-executed and valuable to the membrane and separations community. In response to the reviewer's suggestions, we have carefully addressed the identified points and incorporated the recommended revisions into the manuscript.

Comment

First is the justification of the use of fluorinated polymer membranes for membrane-based separations. Because of the extremely negative environmental impact of these forever chemicals (fluorinated compounds), the use of fluorinated polymers has been banned/restricted in the US and many developed nations, and there have been active research to isolate and destroy these fluorinated compounds.

The motivation to specifically use fluorinated polymers for separations (although the recent restrictions) need to be included in the introduction.

Response: We thank the reviewer for raising an important and timely concern regarding the environmental impact of fluorinated compounds. We fully acknowledge the growing regulatory restrictions and public health concerns associated with persistent fluorinated chemicals. Nonetheless, certain fluorinated polymers continue to play a critical role in membrane-based separations due to their exceptional chemical resistance, thermal stability, and anti-plasticization properties, which remain challenging to replicate with non-fluorinated alternatives [B. Kraftschik et al., *J. Membr. Sci.*, 2013, 428, 608-619, Y. Ren et al., *Science*, 2025, 387, 208-214, B. Liang et al., *Nat. Chem.*, 2018, 10, 961-967]. Interestingly, some fluorinated polymers are even employed in the removal of fluorinated contaminants due to their selective

separation characteristics [Z. Yang et al., *Nat. Comm.*, 2024, 15, 8269]. In our work, the fluorinated polymer is not used as a final membrane material, but rather as a reactive precursor that undergoes thermal conversion to form a microporous, cross-linked network with significantly reduced fluorine contents. This strategy enables the use of fluorinated aromatics as functional intermediates while aiming to minimize their environmental persistence in the final product. In light of the reviewer's suggestion, we have revised the Introduction to clarify this motivation and to acknowledge current regulatory challenges. We are also actively exploring next-generation polymer systems that can achieve high performance with a reduced environmental footprint.

In the revised manuscript Page 4, Line 75-79 and Line 85-95:

“...More recent studies have also reported success in improving plasticization resistance by incorporating rigid fluorinated side chains^[5] or triptycene units^[6]. More recent studies have also reported success in improving plasticization resistance by incorporating bulky and rigid moieties, such as fluorinated side chains¹³ or triptycene units¹⁴. Despite increasing regulatory scrutiny over the persistence of fluorinated compounds, certain fluorinated polymers remain indispensable in membrane research due to their superior chemical resistance and thermal stability, which are difficult to achieve with non-fluorinated analogs.

This study presents an innovative and effective strategy for fabricating highly permeable, plasticization-resistant microporous polymer membranes derived from fluorinated aromatic glassy polymers, termed extrinsically microporous polymeric membranes (EMPMs). The approach leverages a thermally activated defluorination mechanism that generates free radical sites, which undergo inter-chain cross-linking, leading to the formation of a microporous polymer network with reduced residual fluorine content (Fig. 1a). Comprehensive thermal and spectroscopic analyses confirm the evolution of fluorinated fragments, the formation of aryl-aryl linkages, and structural rearrangements that give rise to a highly interconnected free volume architecture. These EMPMs demonstrate exceptional gas permeability and CO₂ separation performance, particularly under sub-ambient conditions, along with outstanding long-term stability and plasticization resistance. Furthermore, the membranes can be readily processed into asymmetric hollow fiber configurations,

highlighting their scalability and practical potential. Overall, this work presents a promising pathway toward high-performance membrane materials, contributing to the advancement of sustainable gas separation technologies.”

Comment

Second, the common issue with thermally crosslinked membrane is mechanical stability. A very brief few sentences were included, but not sufficient. Based on TR literature and color change (Fig 1), mechanical stabilities and film weakness should be further discussed.

Response: We thank the reviewer for this insightful and important comment. We fully agree that mechanical stability is a critical consideration for thermally cross-linked membranes, particularly in relation to the structural evolution that accompanies high-temperature treatment. As noted in our response to reviewer 1 (Comment 2), we conducted tensile testing to quantitatively assess the mechanical performance of the membranes. The *p*TPPFA thermally treated at 450 °C exhibits enhanced mechanical robustness, as evidenced by increased tensile strength and a significantly higher Young's modulus, along with a reduced elongation at break. These findings are consistent with the formation of a rigid, cross-linked network and are summarized in **Figure R3** and **Table R1**.

In addition, we have expanded the discussion of the observed color change shown in **Fig. 1c**, which qualitatively reflects the progression of structural changes during thermal treatment. This visual transformation, from a lighter to a darker hue, is consistent with trends reported in thermally rearranged (TR) polymer systems, where increased aromatic conjugation and network densification lead to stiffening, and in some cases, embrittlement [H.B. Park et al., *Science*, 2007, 318, 254–258, H.B. Park et al., *J. Membr. Sci.*, 2010, 359, 11–24, S. Li et al., *J. Membr. Sci.*, 2013, 434, 137–147, J. Zhang et al., *J. Membr. Sci.*, 2023, 688, 122115]. Despite the observed reduction in ductility, the *p*TPPFA 450 °C membranes remain free-standing and mechanically stable under both routine handling and rigorous testing conditions. Importantly, they maintain structural integrity under high-pressure mixed-gas separation conditions (up to 850 psi, CO₂/CH₄), further confirming their mechanical durability in practical applications.

These points have now been incorporated into the main text to more thoroughly address the mechanical implications of thermal cross-linking, particularly in the context of relevant TR polymer literature.

In the revised manuscript Page 9, Line 197-214:

“The thermally treated membranes (thickness: 60 ± 5 μm) retained their structural integrity.

Nano-indentation analysis (Supplementary Fig. 14) revealed increased hardness and reduced modulus, indicating enhanced mechanical robustness consistent with cross-linked polymer behavior. In addition, tensile stress-strain measurements (Supplementary Fig. 15 and Table S2) confirm that *p*TPPFA membranes treated at 450 °C exhibit significantly improved tensile strength (68.9 ± 2.1 MPa) and Young's modulus (3.93 ± 0.46 GPa), accompanied by reduced elongation at break (2.39 ± 0.27 %), illustrating a typical trade-off between stiffness and ductility^{12,24}. As shown in Fig. 1c, the progressive color change from light to dark serves as a qualitative indicator of increasing aromaticity and densification, consistent with phenomena commonly reported in TR polymer systems²⁵. While such transitions are often associated with increased rigidity and potential brittleness, the *p*TPPFA 450 °C membranes retain excellent flexibility, as demonstrated in Supplementary Videos 1 and 2, especially in contrast to CMS membranes carbonized at 675 °C. Moreover, the membranes maintain integrity under CO₂/CH₄ feed conditions at pressures as high as 850 psi, further confirming their suitability for practical gas separation applications, which will be discussed in more detail later."

In the revised Supplementary Note 1:

"Tensile tests were carried out using a Shimadzu Autograph AGS-X precision universal testing machine equipped with a 500 N load cell. Membranes with a thickness of 30-50 μm were cut into regular specimens with an effective length of approximately 20 mm and a width of approximately 8 mm. Young's modulus was determined from the initial slope of the stress-strain curve. Tensile strength at break and elongation at break were obtained by averaging the results from three independent tests performed on separate membrane specimens."

Comment

A minor comment is, to be consistent with chemical structure nomenclature (especially with subscript, e.g., CH₄, O₂, etc.)

Response: We appreciate the reviewer's attention to detail. We have carefully reviewed the manuscript to ensure consistent and correct use of chemical nomenclature, including proper formatting of chemical formulas with appropriate subscripts (e.g., CH₄, O₂).

Reply to the comments by Reviewer # 3

In this work, authors synthesized a microporous membrane with cross-linked structure through high-temperature defluorination, which exhibits gas separation performance at low temperatures.

However, the results are not solid to support the hypothesis or conclusion. The following points need to be considered.

Comment

1. The accompanying video can only show the soft behavior of obtain material. How about other mechanical properties (for example, toughness, strength)? Authors should add more videos to support other important mechanical properties.

Response: We thank the reviewer for the valuable comment. We agree that the supplementary videos provide only qualitative insight into the membrane's flexibility. To address this, we performed comprehensive mechanical characterization using tensile stress-strain measurements. This was also noted in our response to reviewer 1 (Comment 2). As shown in **Figure R3**, the thermally treated *p*TPPFA membranes exhibit significantly higher tensile strength and Young's modulus, along with reduced elongation at break, compared to the pristine membrane. These findings confirm the formation of a stiffer, mechanically robust cross-linked structure and are summarized in **Figure R3** and **Table R1**.

The Supplementary Videos 1 and 2 were originally included to qualitatively demonstrate the membranes' flexibility and absence of brittleness, this visual evidence is now reinforced by quantitative mechanical data. Together, these complementary results provide a comprehensive assessment of the membrane's mechanical integrity. We have revised both the manuscript and Supplementary Information to reflect this clarification.

In the revised manuscript Page 9, Line 197-214:

"The thermally treated membranes (thickness: $60 \pm 5 \mu\text{m}$) retained their structural integrity. Nano-indentation analysis (Supplementary Fig. 14) revealed increased hardness and reduced modulus, indicating enhanced mechanical robustness consistent with cross-linked polymer behavior. **In addition, tensile stress-strain measurements (Supplementary Fig. 15 and Table S2)**

confirm that *p*TPPFA membranes treated at 450 °C exhibit significantly improved tensile strength (68.9 ± 2.1 MPa) and Young's modulus (3.93 ± 0.46 GPa), accompanied by reduced elongation at break (2.39 ± 0.27 %), illustrating a typical trade-off between stiffness and ductility^{12,24}. As shown in Fig. 1c, the progressive color change from light to dark serves as a qualitative indicator of increasing aromaticity and densification, consistent with phenomena commonly reported in TR polymer systems²⁵. While such transitions are often associated with increased rigidity and potential brittleness, the *p*TPPFA 450 °C membranes retain excellent flexibility, as demonstrated in Supplementary Videos 1 and 2, especially in contrast to CMS membranes carbonized at 675 °C. Moreover, the membranes maintain integrity under CO₂/CH₄ feed conditions at pressures as high as 850 psi, further confirming their suitability for practical gas separation applications, which will be discussed in more detail later."

In the revised Supplementary Note 1:

"Tensile tests were carried out using a Shimadzu Autograph AGS-X precision universal testing machine equipped with a 500 N load cell. Membranes with a thickness of 30-50 μm were cut into regular specimens with an effective length of approximately 20 mm and a width of approximately 8 mm. Young's modulus was determined from the initial slope of the stress-strain curve. Tensile strength at break and elongation at break were obtained by averaging the results from three independent tests performed on separate membrane specimens."

Comment

2. Structure analyses should be carefully conducted. For example, authors mentioned that “the peak at 115 ppm is due to the newly formed aromatic C-C linkages in the biphenyl moiety” (Fig. 2e). However, the similar peak can also be found in the non-heat-treated membrane (*p*TPPFA).

Response: We thank the reviewer for the careful observation. It is indeed correct that a weak peak near 115 ppm is already present in the solid-state ^{13}C NMR spectrum of the pristine *p*TPPFA membrane. However, this peak becomes significantly more intense and better resolved after thermal treatment at 450 °C, as shown in **Figure R5**.

Figure R5. Solid-state ^{13}C CP/MAS NMR spectra of *p*TPPFA before and after thermal treatment. A peak at 115 ppm, attributed to aryl-aryl ($\text{sp}^2\text{-sp}^2$) carbon linkages, is present in both the pristine and thermally treated membranes. However, its increased intensity and sharper resolution in the *p*TPPFA 450 °C sample suggest the formation of extended biphenyl cross-links via thermal defluorination. The inset highlights this spectral region. While the weak signal in the pristine membrane may originate from intrinsic biphenyl or phenyl units in the polymer backbone, the pronounced enhancement after thermal treatment

indicates the formation of additional C-C aromatic linkages. This interpretation is now clarified in the revised manuscript.

In revised manuscript Page 10, Line 231-245:

“This may indicate the formation of new carbon-carbon bonds following defluorination by thermal activation. Furthermore, the solid-state cross-polarization/magic-angle spinning (CP/MAS) ^{13}C NMR spectra of thermally treated *p*TPPFA membranes revealed a predominantly aromatic structure. Peaks at 126 ppm and 139 ppm correspond to the terphenyl backbone (**Fig. 2e**). For *p*TPPFA 450°C, a distinct peak at 115 ppm is assigned to newly formed aromatic C-C linkages, consistent with biphenyl-type structures. Although a weak signal is also observed at this chemical shift in the pristine *p*TPPFA membrane, its substantially increased intensity and sharper profile in the thermally treated derivatives strongly support its assignment to cross-linking-related structural evolution. Additionally, a small peak at 155 ppm is attributed to ortho-positioned sp^2 carbon atoms in the residual fluorinated biphenyl units. The increased intensity of the peak at 132 ppm further suggests a higher concentration of aromatic C-C linkages in the *p*TPPFA 450°C compared to *p*TPPFA 400°C, indicating progressive structural transformation toward a more densely cross-linked network.”

Comment

3. The authors mentioned that “terphenyl block bonding primarily occurs at the meta position.” (Fig.2g), but the reviewer did not find supporting evidence in the literature (ref. 28) and believes that it is more likely to occur in para position.

Response: We thank the reviewer for this important observation. We agree that *para*-substitution is generally more favorable in terms of sterics and resonance stabilization perspectives. Upon reflection, we acknowledge that the original sentence may have overstated the assignment and that Reference 28 was misplaced, as it does not directly address terphenyl systems or confirm the bonding configuration relevant to our study.

To address this concern, we have revised the corresponding sentence in the manuscript to adopt a more cautious and evidence-based interpretation, grounded in our IR spectral data rather than speculative structural assumptions.

In revised manuscript Page 11, Line 252-256:

“Additionally, a peak at 886 cm^{-1} , commonly associated with *meta*-substituted phenyl groups²⁸, suggests that some newly formed aromatic C-C linkages may involve *meta*-connected units. While *para*-substitution is generally more favorable, our results does not rule out the coexistence of both *meta* and *para* linkages in the thermally cross-linked network.”

This revised wording avoids overgeneralization and instead presents the possibility of *meta*-linkages based on the appearance of the 886 cm^{-1} band, which has been associated with *meta*-substitution in prior IR assignments. Furthermore, we have revised **Fig. 2g** to illustrate the proposed structural transformations of *p*TPPFA upon thermal treatment (350–500 °C), indicating that both *para* and *meta*-linkages may form.

Comment

4. To comprehensively evaluate the membrane's durability and aging resistance, it is necessary to include graphs obtained under long-term continuous operating conditions. For example, a CO₂ permeability-time diagram spanning 72 hours of uninterrupted operation could serve as an indicator of the membrane's stability and performance consistency.

Response: We appreciate the reviewer's insightful suggestion regarding the evaluation of membrane durability under long-term continuous operation. In response, we performed an extended 96 hour mixed-gas permeation test to directly assess operational stability (**Figure R6**). This test was conducted using a 50/50 mol% CO₂/CH₄ mixture at 35 °C and 2 bar total feed pressure, beginning 24 hours after membrane fabrication. Over the course of four consecutive days, the membrane maintained stable performance, with CO₂ permeability decreasing by only 7.14% on Day 5 compared to the initial value on Day 1 (from 10,833 Barrer to 10,059 Barrer).

Figure R6. Long-term mixed-gas permeation performance of pTPPFA 450 °C membrane over 96 hours of continuous operation at 2 bar and 35 °C.

These results confirm the excellent operational durability and reinforce its suitability for realistic, long-term gas separation applications. We have updated the Supplementary

Information and included a corresponding discussion in the revised manuscript.

In revised manuscript Page 18, Line 427-443:

“The *p*TPPFA 450 °C membrane demonstrates remarkable resistance to aging, maintaining a CO₂ permeability above 4,000 Barrer even after 523 days (or 10,000 h) of vacuum storage (Fig. 4e). Physical aging, which typically reduces permeability in PIMs due to polymer chain densification and loss of free volume, is substantially suppressed in this system. The aged *p*TPPFA 450 °C membrane still exhibits competitive CO₂ permeability compared to state-of-the-art membranes⁴⁴. This long-term durability is attributed to the cross-linked network, which restricts chain mobility, mitigates aging, and preserves high CO₂ permeability over prolonged periods. In addition to long-term shelf stability, the membrane also displayed outstanding operational durability under continuous mixed-gas test. A 96 h permeation test using a 50/50 mol% CO₂/CH₄ mixture at 35 °C and 2 bar resulted in only a 7.14% decline in CO₂ permeability, indicating its strong resistance to operational physical aging (Supplementary Fig. 24). Lai *et al.* recently reported selectivity enhancement in aged 3D network ladder polymers⁴⁵. While the EMPM in this study exhibited only a minimal increase in selectivity after aging, selectivity might be improved by introducing kinked monomers during polymer synthesis.”

In the revised Supplementary Information:

“Supplementary Fig. 24. Long-term mixed-gas permeation performance of *p*TPPFA 450 °C membrane over 96 hours of continuous operation. The membrane was tested using a 50/50 mol% CO₂/CH₄ mixture at 2 bar and 35 °C. Both CO₂ and CH₄ permeabilities remained stable, confirming excellent operational stability and resistance to physical aging.”

Comment

5. This material could work at low temperature and high pressures. However, this combination could be a drawback in practical applications due to the low efficiency and high requirements of operations. Authors should add more discussions to support the interest of this contribution.

Response:

We thank the reviewer for raising this important point. We agree that operating processes at both low temperature and high pressure can present practical limitations due to increased energy requirements and the risk of gas condensation. However, in this study, our membrane was evaluated under low-temperature and high-pressure conditions independently, not simultaneously, to assess its robustness across a wide range of industrially relevant scenarios. The motivation for exploring sub-ambient temperature performance stems from recent advancements in “chilled membrane” processes for CO₂ capture and air separation. These processes can enhance selectivity at lower temperatures, potentially offsetting permeability losses and improving overall cost-effectiveness [E. Esposito et al., *J. Membr. Sci.*, 2019, 582, 402]. Typically, cross-linked and microporous polymers experience significant declines in CO₂ permeability at sub-ambient conditions [Lin et al., *Science*, 2006, 311, 57613, Ji et al., *J. Membr. Sci.*, 2021, 623, 119091]. In contrast, *p*TPPFA 450 °C exhibits only a modest reduction in CO₂ permeability, from 12,000 Barrer at 35 °C to 10,324 Barrer at -20 °C, accompanied by significant increases in CO₂/N₂ and CO₂/CH₄ selectivities of 46 and 61, respectively, at -20 °C. We acknowledge the importance of evaluating techno-economic feasibility under combined low-temperature and high-pressure operation, as the reviewer suggested. However, a detailed process simulation and cost analysis under various industrial scenarios is beyond the current scope of this study. We agree this represents an important direction for future research and plan to investigate it systematically in follow-up studies.

In revised manuscript Page 17, Line 402-420:

“The “cold membrane process” has gained attention as a promising approach for energy-

efficient CO₂ capture, particularly for biogas upgrading and post-combustion gas separation^{40,41}. For these processes to be viable, membranes must combine high intrinsic selectivity with sufficient permeability. As shown in **Fig. 4c** and **4d**, the *p*TPPFA 450°C membrane demonstrates impressive CO₂ separation performance, surpassing the 2019 upper bound for ultrapermeable benzotriptycene-based polymers⁴², making it an attractive candidate for the *cold membrane process*. At 35°C, *p*TPPFA 450°C exhibits moderate CO₂/N₂ and CO₂/CH₄ selectivities of 13 and 15, respectively. However, lowering the temperature to -20°C enhances CO₂ adsorption and restricts the diffusion of larger N₂ and CH₄ molecules, boosting selectivities to 46 and 61, respectively. This temperature-dependent improvement aligns with findings by Wenhui Ji et al., who observed significant improvements in PIM-1 performance at sub-ambient temperatures, with CO₂/N₂ selectivity rising from 21.1 at 35°C to 55.6 at -20°C, despite a 48.7% decrease in CO₂ permeability at -20°C. Notably, the CO₂ permeability of *p*TPPFA 450°C remains relatively stable (10,324 Barrer) even at -20°C, resulting in a remarkable increase in separation performance beyond the 2019 upper bound. This unique combination of high selectivity and permeability under sub-ambient conditions positions the EMPM as a highly promising candidate for energy-efficient CO₂ separation in industrial applications.”

Comment

6. It is well-known that C-F bond is among the strongest covalent bond. Authors hypothesized that the formation of C-C bonds took place by breaking the C-F bonds. It is a wired design based on the energetic reason. Authors should discuss literatures and design related tests to support the structure change.

Response: We thank the reviewer for this thoughtful and important comment. It is indeed well established that C-F bonds are among the strongest covalent bonds. However, our proposed mechanism does not rely on isolated homolytic cleavage of individual C-F bonds under mild conditions. Rather, it is grounded in thermodynamic and experimental evidence indicating that highly fluorinated aromatic systems can become destabilized under elevated temperatures, enabling defluorination and aryl radical formation. For example, the classic thermodynamic study by Cox et al. (*Trans. Faraday Soc.*, 1964, 60, 653–665) revealed that the heat of formation of hexafluorobenzene is approximately 159 kJ mol⁻¹ (38 kcal mol⁻¹) less negative than predicted by additive bond energy models. This deviation suggests that electronic repulsion (e.g., dipole-dipole repulsions between C-F bonds) and resonance destabilization reduce the overall thermodynamic stability of the fluorinated aromatic system, thereby lowering the effective energy barrier for defluorination under thermal conditions. Further experimental support comes from Sato et al. (*ACS Nano*, 2008, 2, 348–356) who demonstrated that fluorinated carbon nanotubes (CNTs) can undergo thermal defluorination, generating reactive sites that enable covalent cross-linking via radical coupling. Their study clearly shows that thermally driven defluorination can result in the formation of mechanically robust, binder-free CNT assemblies. Although their system involves CNTs, the underlying radical-driven mechanism is directly applicable to fluorinated aromatic polymers, such as those studied in our work.

Based on these literature reports, our design strategy leverages thermally activated defluorination to generate reactive aryl radicals, which then recombine to form new C-C bonds, resulting in a cross-linked, microporous network. To support this mechanism, we have conducted a series of spectroscopic and thermal analyses—including solid-state ¹³C NMR, FT-IR, DSC, and TG-MS—which together provide strong evidence for structural evolution

consistent with aryl radical formation and cross-linking.

We appreciate the reviewer's comment, which prompted us to expand the mechanistic discussion and integrate relevant literature support in the revised manuscript.

In revised manuscript Page 8, Line 165-171:

“Despite the higher bond dissociation energy (BDE) of C-F bonds compared to C-C bonds (485 vs. 348 kJ mol⁻¹), TG-MS results suggest that pentafluorophenyl fluorine atoms may form highly reactive radicals upon thermal activation¹⁴. This observation aligns with earlier thermodynamic work by Cox¹⁸ et al., who reported that the heat of formation of hexafluorobenzene deviates by approximately 159 kJ mol⁻¹ (38 kcal mol⁻¹) from values predicted by bond additivity, indicating that electronic repulsion within fully fluorinated aromatic ring reduces its thermodynamic stability. Such destabilization facilitates defluorination and may promote the formation of aryl radicals, ultimately driving thermally induced cross-linking.”

Comment

7. Fig. 2b TG-MS test results show that the C₆F₅ signal appears at a temperature close to or even lower than that of the F signal, suggesting that C₆F₅ may be more easily removed than F, which is contrary to the authors' idea.

Response: We thank the reviewer for this insightful comment. While it may initially appear counterintuitive that the C₆F₅⁺ fragment (m/z = 168) is detected at a temperature close to—or slightly lower than—that of the F⁻ ion (m/z = 19), this observation can be rationalized by considering the complexity of thermal degradation and defluorination pathways in fluorinated polymers.

First, the early evolution of C₆F₅⁺ is likely attributed to the cleavage of thermally labile moieties such as side chains, oligomeric segments, or chain-end groups, rather than decomposition of the main cross-linked backbone. These peripheral components are more susceptible to thermal degradation and may evolve at lower temperatures. Additionally, C₆F₅⁺ fragment is known to be a relatively stable gas-phase species and may ionize more efficiently than F⁻ in the TG-MS system, leading to earlier or sharper detection (**Figure R7**).

Figure R7. (a) Simultaneous thermogravimetric analysis (TGA) coupled with mass spectroscopy (MS) of pTPPFA under N₂ atmosphere. The gray line represents the TGA weight loss profile, while the red dashed line corresponds to the first derivative of TG (DTG). Ion currents for selected mass-to-charge (m/z) ratios are plotted on the blue right y-axis. (b) Normalized ion current of released F (m/z=19) and C₆F₅ (m/z=168).

Importantly, our proposed mechanism does not assume a strictly sequential or uniform loss of individual fluorine atoms. Rather, we describe defluorination as a chemically complex, multi-pathway process involving: direct C-F bond cleavage, partial elimination of fluorinated

aryl groups (e.g., C₆F₅), and formation of aryl radicals that recombine to form C-C cross-links. This interpretation is supported by complementary evidence from ¹⁹F MAS NMR and XPS (**Fig. 2d and Supplementary Fig. 16**), which demonstrate a reduction in C-F signal intensity and confirm that partial defluorination occurs as early as 400 °C. However, as noted in our response to reviewer 1 (Comment 1), extended thermal treatment at 400 °C—even with evolution of C₆F₅⁺—does not lead to the formation of microporosity. This finding strongly suggests that initial defluorination of C₆F₅⁺ evolution alone are not sufficient to induce EMPM formation. Instead, a higher degree of defluorination, achievable only at temperatures of 450 °C or higher, is required to trigger extensive aryl radical generation, C-C coupling, and the development of a robust cross-linked network. Given the structural complexity of thermally treated fluorinated polymers, we propose that the cross-linking process involves multiple, potentially concurrent, pathways, including radical-mediated C-C coupling and partial backbone rearrangements. Therefore, the early detection of C₆F₅⁺ is not contradictory to our proposed mechanism; rather, it reflects the heterogeneous and stepwise nature of defluorination and structural evolution during thermal treatment.

Finally, in response to reviewer 3's Comment 3,6, and 7, we have also revised the chemical structures and proposed cross-linking pathway in **Fig. 1a** and **Fig. 2g** to better reflect this mechanistic complexity.

In the original manuscript: Fig. 1a

In the revised manuscript: Fig. 1a

In the original manuscript: Fig. 2g

In the revised manuscript: Fig. 2g

Comment

8. The sizes of tested gaseous molecules are at nanometer scale. But the SEM and related test are at micrometer scale, thousand times larger than the guest molecules. It is necessary to add more characterizations at a relevant scale to support this work.

Response: We thank the reviewer for this important comment. We fully agree that SEM and other micrometer-scale imaging techniques cannot directly capture the sub-nanometer-scale features that govern molecular-level gas transport. To address this, we have employed complementary characterization and analytical approaches relevant to gas-scale diffusion. Specifically, we analyzed the CO₂ permeability of both pristine *p*TPPFA and thermally treated *p*TPPFA 450 °C membranes as a function of pressure, as presented in the original manuscript. The pressure-dependent permeability behavior was interpreted using the dual-mode transport model, which is widely used to describe gas sorption and diffusion in glassy polymers. This model provides insight into how small gas molecules such as CO₂ (with a kinetic diameter of ~3.3 Å) diffuse through the microporous polymer matrix. Our results indicate that the thermally cross-linked structure retains sufficient free volume to facilitate CO₂ diffusion across a wide pressure range, while maintaining high resistance to plasticization. In addition, 77 K N₂ adsorption and XRD measurements further support the presence of sub-nanometer-scale microporosity, confirming that the EMPMs possess structural features directly responsible for their selective gas transport properties.

If the SEM image mentioned by the reviewer refers to **Fig. 5**, we would like to clarify that it was included to confirm the presence and thickness of the coated dense selective layer in the asymmetric EMP hollow fiber membrane. This cross-sectional SEM image serves to validate the overall membrane architecture and uniformity, rather than to characterize molecular-scale pore structures.

Reviewer 1, attachment 1

This work summarizes the development of cross-linked membranes modified extrinsically microporous polymer membranes (EMPMs) as a novel class of microporous membranes, fabricated from perfluorinated aryl-ether-free aromatic polymers via defluorination-induced thermal cross-linking. Interesting trends in the gas selectivity, the separation performance is above the current upper bound lines for single and mixed-gas pairs. While the scope of characterization is comprehensive, there are some questions that need to be addressed to understand the underlying mechanism of what is occurring in these membranes.

1. According to the results of the characterization in the paper, the mechanism of pTPPFA membrane is carried out according to the route given in the paper when soaked in 450 °C, what will be the result when the soaked time is prolonged at a low temperature, such as 400 °C or 380 °C for 5-10 h, which will show the same results or the formation of CMS membrane.
2. The author points out the nano-indentation results (Supplementary Fig. 14) showed increased hardness and reduced modulus, indicating enhanced mechanical robustness typical of cross-linked polymers, what about mechanical property (strain-stress).
3. To explain the mechanism more clearly, is it possible to use TG-MS or TG-IR.
4. Is pTPPFA can be made into Asymmetric pTPPFA hollow fiber membranes (HFMs), and pTPTFA can not?
5. Do author change the current structure, if put F or CF₃ in the triphenyl will have the current mechanism?
6. There are some typographical errors that need to be corrected, such as the lower corners or Spaces (400 °C), and so on.